# Molecular and electrophysiological features of spinocerebellar ataxia type seven in induced pluripotent stem cells

Richard J. Burman [1,2,3☯]*, Lauren M. Watson[4,5☯], Danielle C. Smith[4,5☯], Joseph V. Raimondo[1,2,5], Robea Ballo[1], Janine Scholefield[6], Sally A. Cowley[7], Matthew J. A. Wood[8], Susan H. Kidson[1,5], Leslie J. Greenberg[4,5]

1 Department of Human Biology, University of Cape Town, South Africa, 2 Neuroscience Institute, University of Cape Town, South Africa, 3 Nuffield Department of Clinical Neurosciences, University of Oxford, Oxford, Oxfordshire, United Kingdom, 4 Department of Pathology, University of Cape Town, South Africa, 5 Institute of Infectious Disease and Molecular Medicine, University of Cape Town, South Africa, 6 Gene Expression & Biophysics Group, Synthetic Biology ERA, CSIR Biosciences, Pretoria, Gauteng, South Africa, 7 Sir William Dunn School of Pathology, University of Oxford, Oxford, Oxfordshire, United Kingdom, 8 Department of Paediatrics, University of Oxford, Oxford, Oxfordshire, United Kingdom

☯ These authors contributed equally to this work.
* richard.burman@ndcn.ox.ac.uk

**Data Availability Statement:** The data underlying the results presented in the study are available from: https://datadryad.org/stash/share/

## Abstract

Spinocerebellar ataxia type 7 (SCA7) is an inherited neurodegenerative disease caused by a polyglutamine repeat expansion in the *ATXN7* gene. Patients with this disease suffer from a degeneration of their cerebellar Purkinje neurons and retinal photoreceptors that result in a progressive ataxia and loss of vision. As with many neurodegenerative diseases, studies of pathogenesis have been hindered by a lack of disease-relevant models. To this end, we have generated induced pluripotent stem cells (iPSCs) from a cohort of SCA7 patients in South Africa. First, we differentiated the SCA7 affected iPSCs into neurons which showed evidence of a transcriptional phenotype affecting components of STAGA (*ATXN7* and *KAT2A*) and the heat shock protein pathway (*DNAJA1* and *HSP70*). We then performed electrophysiology on the SCA7 iPSC-derived neurons and found that these cells show features of functional aberrations. Lastly, we were able to differentiate the SCA7 iPSCs into retinal photoreceptors that also showed similar transcriptional aberrations to the SCA7 neurons. Our findings give technical insights on how iPSC-derived neurons and photoreceptors can be derived from SCA7 patients and demonstrate that these cells express molecular and electrophysiological differences that may be indicative of impaired neuronal health. We hope that these findings will contribute towards the ongoing efforts to establish the cell-derived models of neurodegenerative diseases that are needed to develop patient-specific treatments.

## Introduction

Spinocerebellar ataxia type 7 (SCA7) is an inherited neurodegenerative disease caused by a CAG repeat expansion in the *ATXN7* gene. Since the translation of this CAG repeat leads to an expanded polyglutamine (polyQ) tract within the resultant protein, SCA7 is classified as a polyQ repeat disorder. Other diseases with a similar pathophysiology include five different SCAs (SCA 1, 2, 3, 6 and 17), as well as Huntington disease, dentatorubral-pallidoluysian atrophy and spinal bulbar muscular atrophy [1]. Clinically, SCA7 patients present with ataxia, dysarthria and visual loss. This is caused by a selective degeneration of cerebellar Purkinje neurons and retinal photoreceptors [2]. Symptoms progressively worsen over a period of 10 to 30 years, leading ultimately to brainstem dysfunction, blindness, physical disability and death.

The mechanism by which a polyQ expansion within the ubiquitously expressed ATXN7 protein leads to the selective degeneration of Purkinje neurons and photoreceptors remains to be fully elucidated. ATXN7 is known to be a component of the mammalian transcription co-activator complex, STAGA (SPT3-TAF9-ADA-GCN5 acetyltransferase) [3]. This protein has been shown to facilitate the interaction between STAGA and the cone-rod homeobox (CRX) transactivator of photoreceptor genes, linking the function of ATXN7 with the retinal phenotype observed in SCA7 patients [4]. In addition, several studies have highlighted the role of transcriptional aberrations in the neuronal cell dysfunction that precedes the onset of disease symptoms [3–7]. These gene expression changes may arise either directly from alterations in transcriptional regulation by mutant ATXN7, or indirectly, as a consequence of a generalised cellular response to the presence of mutant ATXN7. More recently, dysfunction in cell metabolism has been proposed as an additional pathogenic mechanism [8, 9].

As with many neurodegenerative conditions, research into the molecular pathogenesis of SCA7 has been hindered by a lack of suitable models of human disease progression. This is particularly relevant in cases where the genomic context of the mutation may have an impact on gene function and might prove useful for therapeutic development. There does, however, seem to be increasing momentum in this area with new models being proposed and refined [10–12].

SCA7 occurs at an unusually high frequency in the South African population as a result of a founder effect in patients of Black African ethnic origin [13]. South African SCA7 patients also display a unique phenomenon–a single nucleotide polymorphism (SNP) within *ATXN7* (rs3774729), which is linked to the mutation in all patients studied to date [14]. Approximately 43% of these individuals are heterozygous for the polymorphism, allowing for allelic discrimination, and providing an ideal target for developing an allele-specific silencing therapy. This haplotype has been shown to extend into other Southern African populations, suggesting that such a therapy may be more widely applicable than was first thought [15]. We have previously demonstrated the efficacy of an allele-specific RNAi treatment in an over-expression cell model of SCA7 [16], as well as in SCA7 patient fibroblasts [17]. Disease-relevant cell lines generated from these patients are thus of vital importance in the understanding of disease pathogenesis and the development of therapies, as they carry the patient's full genomic sequence, including SNPs which may be used as targets for gene silencing.

The generation of induced pluripotent stem cells (iPSCs) involves reprogramming somatic cells to a pluripotent state by means of viral transduction with the pluripotency genes *OCT4*, *SOX2*, *KLF4* and *c-MYC* [18, 19]. Importantly, iPSCs can then be differentiated into any tissue of the body through treatment with specific growth factors. This makes iPSCs a useful starting point for the generation of disease-relevant cell models, particularly in neurodegenerative diseases, where primary CNS cultures may only be obtained using invasive methods.

In this study, we have generated and characterised iPSC lines from two South African SCA7 patients and an unaffected, related control. We were then able to differentiate these iPSCs into cells expressing markers associated with retinal photoreceptors, neural progenitor cells (NPCs) and neurons. Thereafter, we have obtained preliminary evidence for a disease phenotype in these cells, in the form of pathogenically relevant gene expression changes and alterations in intrinsic neuronal properties. Our findings indicate that these cells may be able to replicate, to some extent, the pathogenesis underlying SCA7.

## Material and methods

### Ethics approval and patient recruitment

Ethics approval for the study was granted by the University of Cape Town (UCT) Faculty of Health Sciences Human Research Ethics Committee (HREC REF. 380/2009 and 434/2011), and was renewed annually, incorporating amendments to the project protocol where necessary. All methods were carried out in accordance with the guidelines approved by the Ethics Committee. Participants were recruited from the Neurogenetics clinic at Groote Schuur Hospital in Cape Town, where they were counselled by a genetic counsellor and a clinical geneticist. Written informed consent was obtained from all participants prior to their enrolment in the study.

### Establishment of primary fibroblast cultures

Primary dermal fibroblast cultures were established from punch skin biopsies [20] taken from the inner forearm of two unrelated SCA7 patients (47Q and 41Q) and an unaffected control individual (CON, sibling of 41Q) who had consented to participate in the study. CAG genotypes and ages at diagnosis and biopsy are shown in S1 Fig.

### Generation and characterisation of patient-derived iPSCs

Dermal fibroblasts were reprogrammed into iPSCs using a replication-defective and persistent Sendai virus vector (SeVdp) containing OCT4, SOX2, KLF4 and c-MYC as previously described [21, 22]. One week after infection, the cells were transferred to mitomycin C-inactivated mouse embryonic fibroblast feeder layers in stem cell medium (KO DMEM, 20% KOSR, 1% NEAA, 50 μM β-mercaptoethanol, glutamax, 10ng/ml bFGF). Half of the medium was refreshed every day and colonies were expanded by manual passaging (dissection of colonies using needles) on feeder layers as previously described by Schwartz et al. [23]. After 10 passages the expression of pluripotency markers OCT4 and TRA-1-60, as well as silencing of the reprogramming SeVdp vector, were confirmed by immunocytochemistry (antibodies listed in S1 and S2 Tables). The expression of selected pluripotency genes (OCT4, SOX2, NANOG) was determined by quantitative real-time reverse transcriptase PCR (qRT-PCR). Genomic integrity was assessed by means of karyotype analysis (G-banding, S2 Fig). To confirm the iPSC lines' capacity to differentiate into the three embryonic germ layers, in vitro differentiation via embryoid body formation was performed as previously described [24].

Following SeVdp-mediated reprogramming, iPSC colonies with the correct morphology (flat, with distinct borders, containing tightly packed cells with a high nucleus-to-cytoplasm ratio) appeared within three to four weeks. These colonies were manually picked and clonally expanded in separate dishes on a feeder layer of inactivated mouse embryonic fibroblasts. Two SCA7 patient iPSC lines and two control lines (representing two affected individuals and a single related, unaffected control) were successfully generated and characterised. Immunocytochemical analysis revealed iPSC colonies with distinct nuclear staining for the pluripotency

transcription factor, *OCT4*, compared to the surrounding feeder fibroblasts (S3A Fig–top panel). The iPSC colonies also stained positive for the embryonic stem cell surface marker TRA-1-60 (S3A Fig–bottom panel). In addition, we confirmed the differentiation capacity of these cells into mesoderm, ectoderm and endoderm (S4 Fig).

Co-staining of all iPSC lines with primary antibodies against OCT4 and the nucleocapsid protein of the virus (anti-NP) showed little or no evidence of NP staining in the OCT4-positive pluripotent cells (S3B Fig–bottom panel), compared to intense cytoplasmic staining in newly infected fibroblasts (S3B Fig–top panel). This indicated that the Sendai virus had been effectively silenced, and that the iPSCs had achieved self-regulating pluripotency. All lines were assessed after passage 8. The expression of pluripotency markers was further confirmed by qRT-PCR. All five iPSC lines expressed high levels of *OCT4*, *SOX2* and *NANOG*, standard markers of pluripotency (S3C–S3E Fig), compared to donor fibroblasts or cells that had been subjected to retinal or neuronal differentiation. The expression levels of *SOX2* and *NANOG* were similar across the five iPSC lines, but lines 41Q and $CON_B$ showed lower levels of *OCT4* expression compared to the remaining three lines (although still significantly higher than differentiated fibroblasts and retinal cells).

## Neural differentiation and characterisation

Differentiation of iPSCs into neural progenitors was performed by treatment of iPSCs with 3μM glycogen synthase kinase 3 (GSK3) inhibitor (CHIR99021) and 2μM TGFβ inhibitor (SB431542) as previously described, with neural progenitors appearing in culture after seven days. For neuronal differentiation, neural progenitors were seeded at a density of 150 000 cells/well onto a Matrigel-coated six-well plate in neural induction medium [25]. After two days, medium was changed to neuronal differentiation medium (DMEM/FBS supplemented with N2, B27, 300ng/ml cyclic AMP (Sigma), 0.2mM ascorbic acid (Sigma), 10ng/ml BDNF (Peprotech) and 10ng/ml GDNF (Peprotech)) and the cells maintained in culture for 14 to 21 days, with neurite outgrowth typically observed after one week of culture. Medium was changed every 2–3 days. Characterisation was performed by immunocytochemistry and qRT-PCR (for antibodies and primers see S1–S3 Tables).

## Electrophysiology

Neurons cultured between 2–3 weeks on glass cover slips (uncoated) were removed from the incubator and rapidly transported to the recording chamber of a Zeiss Axioskop Upright Microscope (Zeiss). Electrophysiological recordings were made in neuronal differentiation medium at room temperature and were restricted to the first 5 hours following cell removal from the incubator environment. Patch pipettes of 13–20 MOhm tip resistance were pulled from filamental borosilicate glass capillaries (2.00 mm outer diameter, 1.58 mm inner diameter, Hilgenberg), using a horizontal puller (Model P-1000, Sutter). The pipettes were filled with an internal solution containing (in mM): K-gluconate (126); KCl (4); Na 2 ATP (4); NaGTP (0.3); Na 2 -phosphocreatinine (10) and HEPES (10). Osmolarity was adjusted to between 290 and 300 mOsM and the pH was adjusted to between 7.38 and 7.42 with KOH. Cells were visualised using a 40x water-immersion objective (Zeiss). Digital images were obtained using a CCD camera (VX55, TILL Photonics). Individual cells were selected for recordings based on a small round or ovoid cell body (diameters, 5–10 μm) and typically two or more extended processes. Recordings were made in current-clamp and voltage-clamp mode using an Axopatch 200B amplifier (Molecular Devices). Data acquisition was performed through an ITC-1600 board (Instrutech) connected to a PC running a custom-written routine (PulseQ) under IGOR Pro (Wavemetrics). Analysis was performed using custom-written scripts in MATLAB

(Mathworks). Statistical analysis was performed using GraphPad Prism (v8, GraphPad Software). Normally distributed data are presented with mean and standard error of the mean (SEM) whilst non-normally distributed data are presented with median and interquartile range (IQR). Appropriate parametric and non-parametric tests were performed including unpaired Students t-test, Mann-Whitney U test and the Kruskal-Wallis test. Significance was defined as $p < 0.05$. For a comparison of means across all cell lines, an ANOVA test was used. Assessment of categorical variables was done using a Chi-Squared test.

## Retinal differentiation and characterisation

Differentiation into retinal photoreceptors was performed as previously described [26], using iPSCs cultured in feeder-free conditions on Matrigel in mTESR$^{TM}$ medium. The cells were dissociated enzymatically and plated onto Matrigel-covered dishes in neural differentiation medium containing N2 and B27 supplements (Life Technologies). After settling for an hour, adhered cells were covered in a 2% Matrigel solution. The following day, and thereafter every second day, the medium was replaced with neural differentiation medium without Matrigel. From day 10 the medium was supplemented with 3nM recombinant SHH (R&D Systems), 50ng/μl acidic fibroblast growth factor (αFGF) (R&D Systems), 10ng/μl basic fibroblast growth factor (bFGF) (Miltenyi), 1mM taurine and 500nM retinoic acid (both Sigma Aldrich). After 30 days of culture characterisation was performed by immunocytochemistry and qRT-PCR (for antibodies and primers, see S1 and S2 Tables). Each patient and control line underwent two rounds of differentiation, with each time point analysed in biological duplicate.

## DNA and RNA isolation, cDNA synthesis

DNA extraction from cultured cells was performed using the QIAGEN DNeasy Blood and Tissue Kit. RNA was isolated from cultured cells using the QIAGEN RNeasy Plus Mini Kit. Synthesis of cDNA was performed using the Applied Biosystems High Capacity cDNA Reverse Transcription Kit (Life Technologies) using 500ng-1μg template RNA.

## Quantitative RT-PCR

qRT-PCR was performed on the BioRad CFX96 Real-Time PCR System, using the Power SYBR Green PCR Master Mix (Applied Biosystems), according to manufacturer's instructions. Primers (S3 Table) were obtained from PrimerDesign Ltd, Integrated DNA Technologies (IDT), or Sigma-Aldrich. Relative quantities of target mRNA were determined using the relative standard curve method [27]. Standard curves were prepared for each primer pair, from serial dilutions of pooled sample cDNA. Universal cycling conditions were used (95°C for 10min, followed by 40 cycles of 95°C for 15 seconds and 60°C for 1min). PCRs were performed in technical triplicate, on at least two biological replicates, and results were analysed using the BioRad CFX Manager software (v3.1). Gene of interest expression was normalised to *β Actin* (*ACTB*) expression in each case. The design and reporting of qRT-PCR experiments aimed to comply with the Minimum Information for publication of Quantitative real-time PCR Experiments guidelines [28, 29] wherever possible. Statistical analysis was performed using the unpaired Students' t-test. Significance was defined as $p < 0.05$.

**CAG repeat length determination.** The length of the disease-causing CAG repeat in *ATXN7* was determined from DNA by means of PCR and automated fluorescent genotyping. The PCR reaction mix consisted of 0.4μM each, forward and reverse primer (S3 Table), 0.6units of GoTaq DNA polymerase (Promega), 100ng of DNA, and Failsafe buffer J (Epicentre Biotechnologies) at a final concentration of 1X, made up to a final reaction volume of 10μl. Cycling conditions were as follows: 95°C for 5min; followed by 30 cycles of 95°C for 30

seconds, 53°C for 6 seconds and 72°C for 40 seconds; and a final elongation step at 72°C for 7min. Automated fluorescent genotyping was performed using the ABI 3130*xl* Genetic Analyzer (Applied Biosystems).

The length of the CAG repeat [$(CAG)_n$] was approximated using the following equation, adapted from (Dorschner et al., 2002): $n$ = (0.3063 x Length of major PCR product in base pairs)– 76.475bp. The major PCR product was defined as the product generating the highest fluorescent peak, as detected using the ABI 3130*xl* Genetic Analyzer.

## Results

### SCA7 iPSC-derived neural progenitor cells show transcriptional disease-associated aberrations

SCA7 patient and control iPSCs generated neural progenitor cells (NPCs) at comparable efficiencies, when cultured in neural induction medium supplemented with SB431542 (TGFβ inhibitor) and CHIR99021 (GSK3 inhibitor), with 100% of the cells expressing the early neural marker Nestin, after five passages (Fig 1A). The cells also stained positive for the disease-causing protein, ATXN7 (Fig 1B), and demonstrated repression of the pluripotency gene *OCT4*, and upregulation of the early neural gene *PAX6* (Fig 1C).

To determine whether any transcriptional differences could be detected between SCA7 patient- and unaffected control-derived cell types, a panel of candidate genes was selected which had previously been shown to be dysregulated in the retinal and cerebellar tissue of SCA7 mouse models and patient lymphoblasts [5, 6, 30, 31]. The panel included the following genes: *ATXN7*, brain expressed, X-linked 1 (*BEX1*), DnaJ (Hsp40) homolog, subfamily A, member 1 (*DNAJA1*), glutamate receptor, ionotropic, AMPA 2 (*GRIA2*), heat shock protein 27 (*HSP27*), heat shock protein 70 (*HSP70*), heat shock protein 105 (*HSP105*), K (lysine) acetyltransferase 2A (*KAT2A*), Phospholipase C, Beta 3 (*PLCB3*) and ubiquitin carboxyl-terminal esterase L1 (*UCHL1*). The expression of these genes was determined by means of qRT-PCR in fibroblasts (S5 Fig) and NPCs (Fig 1D).

In contrast to the fibroblasts and undifferentiated iPSCs, two genes were downregulated in SCA7 patient iPSC-derived NPCs when compared to the unaffected control lines. These included the disease-causing gene *ATXN7* ($p$ = 0.018 in NPCs); and *KAT2A*, encoding GCN5, the histone acetyltransferase (HAT) component of the STAGA transcription coactivator complex ($p$ = 0.003 in NPCs). In addition, NPC-specific alterations in expression were identified in the heat shock protein genes DNAJA1 ($p$ = 0.04) and HSP70 ($p$ = 0.04), both of which were found to be downregulated in SCA7-patient derived cells. Lastly, BEX1, an interactor of the p75 neurotrophin receptor, which regulates neurotrophin signalling and neuronal differentiation, was found to be downregulated in the SCA7 NPCs ($p$ = 0.03).

In order to confirm the size of the *ATXN7* CAG repeat alleles in mRNA from patient- and control-derived fibroblasts, iPSCs and NPCs an RT-PCR-based assay was performed. The results were visualised on an agarose gel (S6 Fig panel A) and confirmed by automated fluorescent genotyping. The length of the CAG repeat did not appear to fluctuate during reprogramming or differentiation, corresponding to previous reports in similar cell lines [32, 33]. This assay also confirmed that both the mutant and wild-type *ATXN7* alleles were expressed by all cell types, and that there were no obvious differences in allele expression in affected or unaffected cells.

After 14 days culture in neuronal differentiation medium containing N2/B27 supplement, cAMP ascorbic acid, BDNF and GDNF [25], both SCA7 patient and control NPCs stained positive for the neuronal marker βIII-Tubulin, and showed robust, diffuse nuclear localisation of the disease-causing protein, ATXN7 (Fig 1E and 1F). A subset of cells (approximately 1.8%)

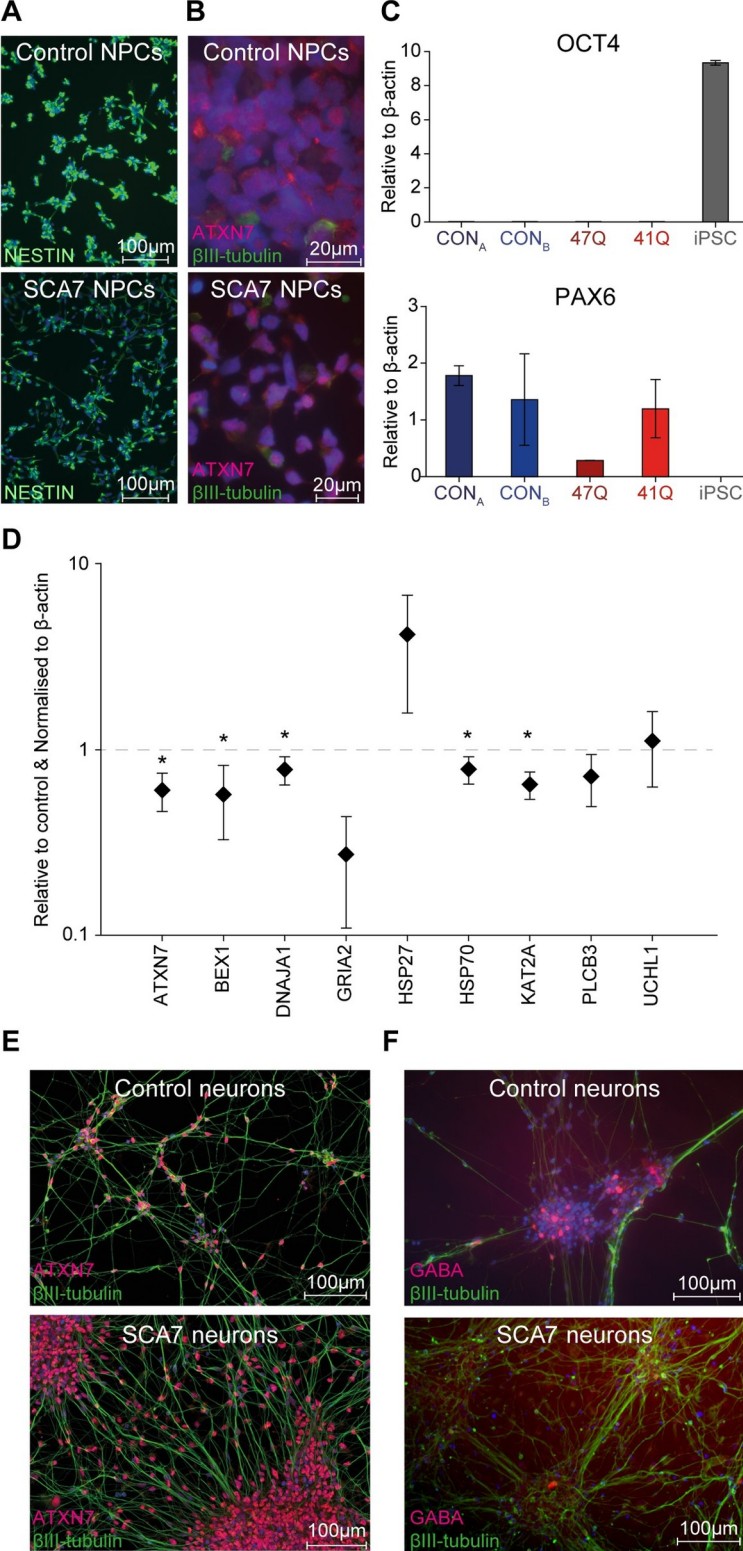

**Fig 1. SCA7 iPSC-derived NPCs show transcriptional aberrations.** (A) Representative images of control (top panel) and SCA7 patient (bottom panel) NPCs, expressing the early neural marker, Nestin (green). (B) High magnification images showing nuclear expression of ATXN7 (red) in control (top) and patient (bottom) NPCs (Green: βIII-Tubulin). (C) qPCR results indicating suppression of *OCT4* (top panel) and upregulation of *PAX6* (bottom panel)

expression in NPCs, compared with the iPSC lines from which they were derived. All levels shown relative to beta ($\beta$)-actin. (D) Expression of 9 selected genes in SCA7 patient NPCs ($n = 2$) shown relative to unaffected control NPCs ($n = 2$). All levels shown relative to $\beta$ actin and unaffected control cells. (E) Control (top panel) and SCA7 NPC-derived neurons (bottom panel) stain positive for $\beta$III-Tubulin (green) and ATXN7 (red) after 14 days of differentiation. (F) Both control and SCA7 patient NPCs produced a small proportion of GABA-positive neurons (red) after 14 days in culture (Green, $\beta$III-Tubulin). $^*p < 0.05$.

stained positive for gamma-aminobutyric acid (GABA), although the proportion of GABAergic neurons did not appear to vary between SCA7 patients and controls (Fig 1E and 1F). No obvious differences in morphology were observed when comparing neurons derived from SCA7 patient iPSCs with those derived from controls. In order to test functional differences between SCA7 neurons and controls, electrophysiological measurements were performed on a subset of these cells.

## SCA7 iPSC-derived neurons demonstrate functional differences that may affect cell excitability

Cells were assayed for physiological properties between 14- and 23-days post induction of neuronal differentiation. Cells were targeted for whole-cell recordings based on their morphological properties. This included a small round or ovoid cell body with diameters between 5–10 $\mu$m and typically two or more extended processes. Following the attainment of a whole-cell patch, current pulses of between 0 and 10 pA were applied. Individual cells displayed four general types of spiking responses: a purely passive (Fig 2A), abortive spike (Fig 2B), single spike (Fig 2C) and recurrent spiking response (Fig 2D). These spiking properties are similar to those observed in acute human fetal brain slices [34], hESC-derived neurons [35–37] and iPSC-derived neurons [38]. A postmitotic neuron matures by inserting voltage-gated channels into its plasma membrane [39]. Therefore, the spiking response of a cell to current injection can be used to determine the maturation stage of a differentiating neuron: passive (least mature) $\rightarrow$ abortive spike $\rightarrow$ single spike $\rightarrow$ recurrent spikes (most mature).

Spiking responses were collected in current-clamp mode from cells derived from four separate iPSC lines: two control lines $CON_A$ ($n = 70$) and $CON_B$ ($n = 42$), and two patient lines 47Q ($n = 44$) and 41Q ($n = 72$). Although the fraction of cells which fell into each spiking response category was significantly dependent on the iPSC line from which the cells were derived ($p < 0.0001$, *Chi-squared test*), no trend could be discerned between control and patient lines (Fig 2E). For example, the control line $CON_A$ had the most mature phenotype with the highest fraction of cells in the single spike and recurrent spiking categories whilst 47Q demonstrated a relatively immature phenotype, with the majority of cells displaying spiking responses falling into the passive category. However, the other lines did not corroborate this difference as the control line $CON_B$ displayed a less mature phenotype than patient line 41Q.

Next, we compared the resting membrane potential (Vm) of cells derived from each cell line. This parameter was significantly dependent on the iPSC line from which the cells were derived (Fig 2F, $p = 0.002$, *ANOVA*). The mean resting membrane potential +/- SEM for the $CON_A$, $CON_B$, 47Q and 41Q cell lines were -57.0 +/- 1.7, -54.7 +/- 3.2, -67.3 +/- 2.7 and -62.5 +/- 2.5 mV respectively.

Next, we performed voltage-clamp recordings of the neurons in order to measure the input resistance as well as the voltage-gated sodium and potassium currents (Fig 3A and 3B). A lower input resistance is associated with neurite outgrowth and increased numbers of ion channels inserted into the plasma membrane during the process of neuronal maturation. Once again, a cell's input resistance was significantly dependent on the cell line to which it belonged (Fig 3C, $p = 0.005$, *ANOVA*). Input resistance +/- SEM was 4987 +/- 421, 5484 +/-

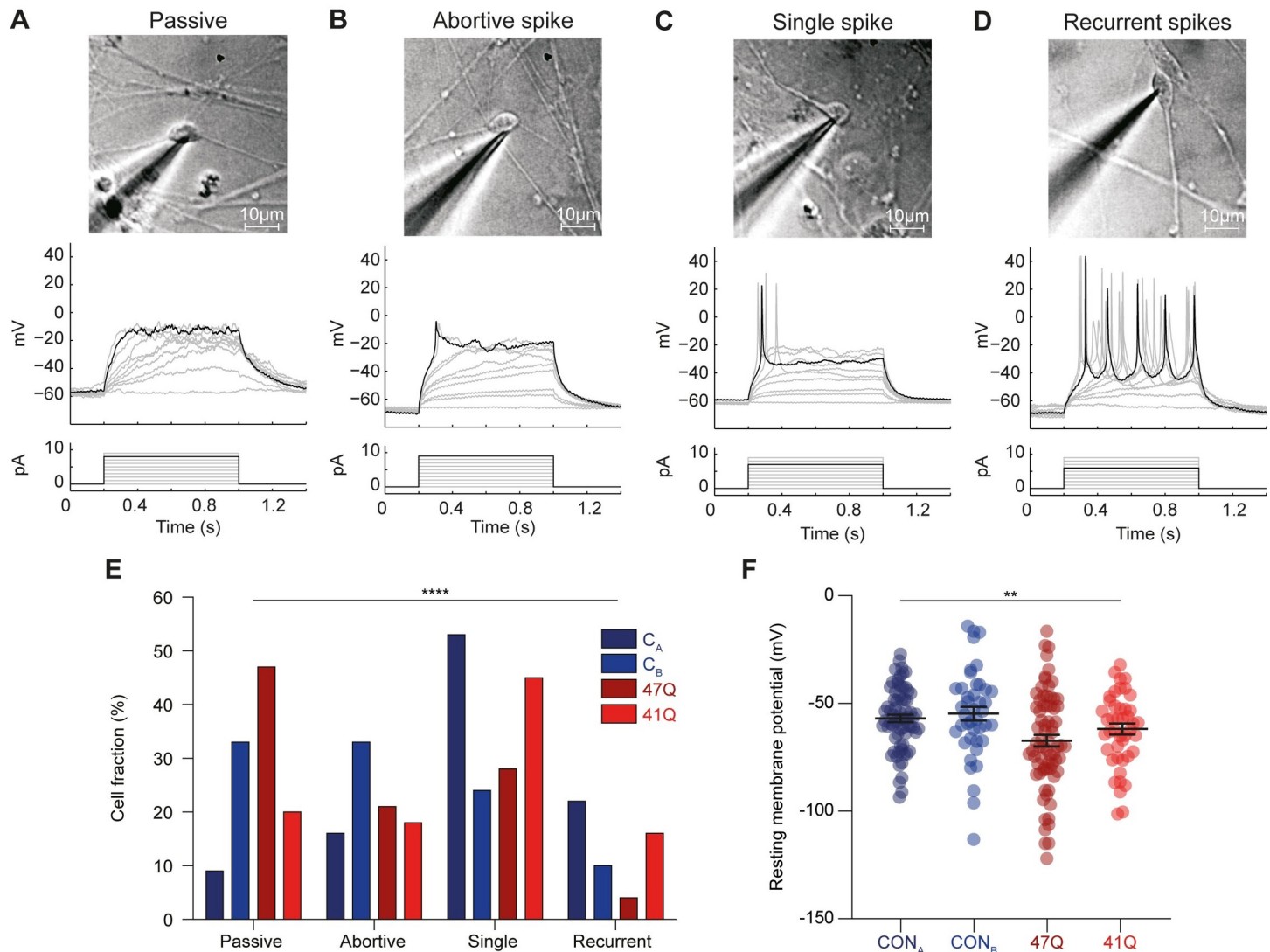

**Fig 2. Patch-clamp analyses of iPSC derived neuronal cultures.** Cells could be divided into four categories based on their spiking responses. (A-D) Differential interference contrast images of cells targeted for patch-clamp recordings (top). Whole-cell recordings in current clamp mode (middle) depict spiking patterns following injection of current (bottom). Cells fell into (A) 'passive', (B) 'abortive spike', (C) 'single spike' and (D) 'recurrent spikes' categories. (E) The fraction of cells which fell into each category for cells derived from two control and two SCA7 patient iPSC lines. Note that although the cells derived from various iPSC lines exhibited different distributions of spiking responses, no trend between control and patient lines was observed. (F) The resting membrane potentials (Vm) of cells varied significantly across each cell line. Error bars denote mean median and IQR. ** $p < 0.01$, *** $p < 0.001$.

243, 5979 +/- 513 and 7094 +/- 455 m$\Omega$ for the $CON_A$, $CON_B$, 47Q and 41Q cell lines respectively.

As one would predict, the spiking properties of neurons are directly correlated with the size of their currents. The maximum size of voltage-gated sodium currents and potassium currents measured in each cell (Max. $I_K$ and Max. $I_{NA}$) was significantly dependent on the particular iPSC line concerned (Fig 3D and 3E, $p < 0.0001$ in both cases, *ANOVA*). The mean Max. $I_K$ +/- SEM was 180.1 +/- 16.5, 176 +/- 10.8, 222.5 +/- 19.3 and 105.7 +/- 8.0 pA for the $CON_A$, $CON_B$, 47Q and 41Q cell lines respectively. The mean Max. $I_{NA}$ +/- SEM was -166.5 +/- 18.6, -277.1 +/- 23.0, -268.0 +/- 30.5 and -108.0 +/- 14.2 pA for the $CON_A$, $CON_B$, 47Q and 41Q cell lines.

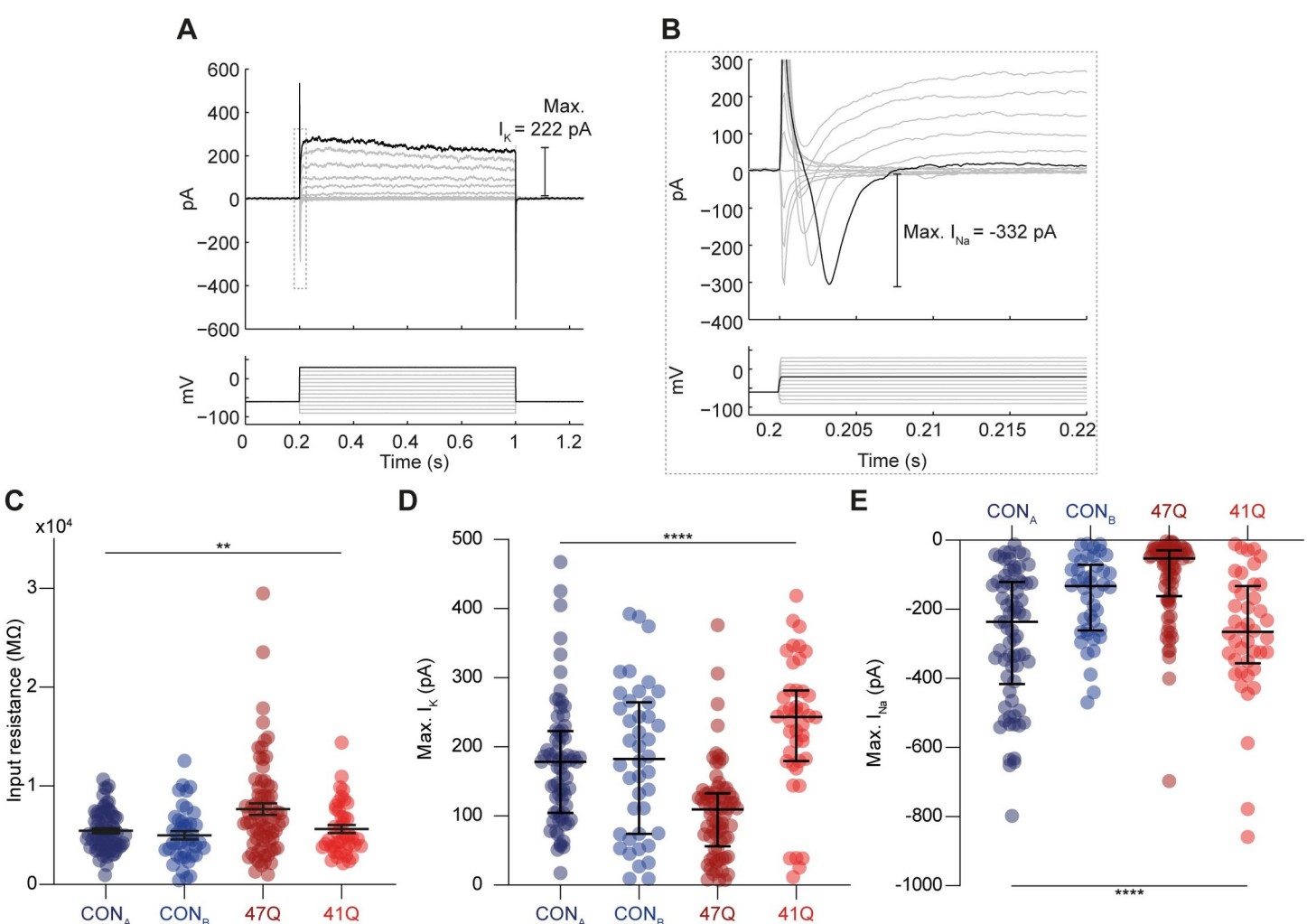

**Fig 3. Sodium and potassium currents in neurons differentiated from control and patient iPSC lines.** (A) Voltage-clamp recording trace. Membrane currents (top) were recorded following 10 mV voltage steps between -90 and 30 mV (bottom). Maximum potassium current ($I_K$) indicated. Dashed gray rectangle indicates the position of voltage-gated sodium currents (enhanced in 'B'). Maximum sodium current ($I_{Na}$) is indicated 'right'. (B) Zoom-in of rectangle in 'A' with the maximum $I_{Na}$ highlighted. (C) Cell input resistance was also significantly varied across cell lines. (D) Population data from the four iPSC lines for maximum voltage-gated $I_K$ and $I_{Na}$ currents showing significant variability across cell lines. Error bars denote median and IQR. *** $p < 0.001$, **** $p < 0.0001$.

## SCA7 iPSC-derived retinal photoreceptors show transcriptional aberrations

For the retinal differentiation of iPSCs, the Matrigel "sandwich" system facilitated a rapid self-organisation and differentiation of the pluripotent stem cells into structures containing cells morphologically indicative of columnar neuroepithelia. These structures lost their integrity from day 4–5, and cells spread into an adherent monolayer. The gradual emergence of cells with a neuronal morphology was observed from day 10 to day 30, particularly in areas of low confluence. Following differentiation of patient and control cells into retinal cells, immunocytochemical analyses were performed on cells at the end of the differentiation period (day 30), to determine whether the cells expressed retinal cell markers. The cells were stained for either the disease-causing protein ATXN7 (Fig 4A) or the retinal cell markers CRX and RCVRN (Fig 4B and 4C). No obvious differences in morphology were observed between SCA7 patient and

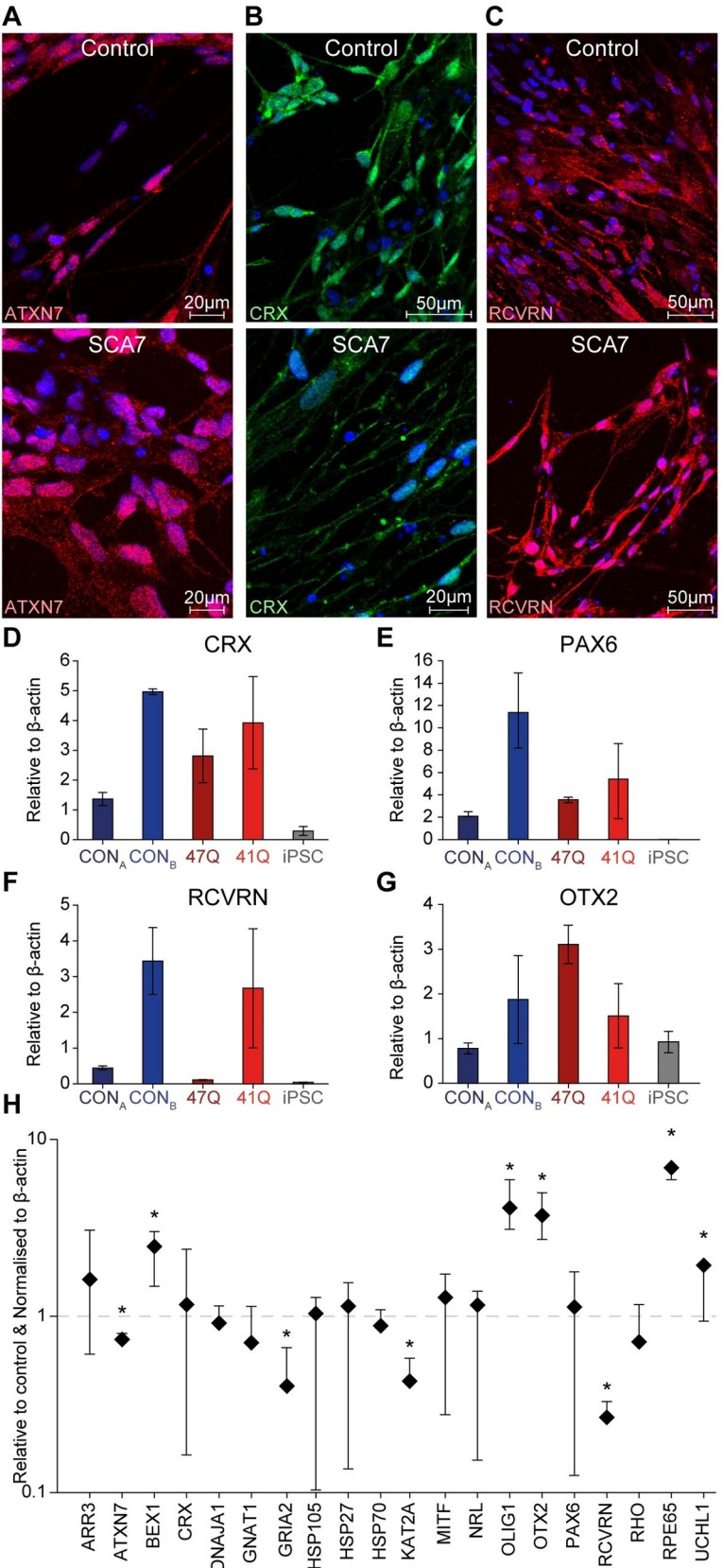

**Fig 4. iPSC-derived retinal cells exhibit aberrant transcriptional changes.** Immunocytochemical analysis of control (top panel, representative images from line $CON_B$) and SCA7 (bottom panel, representative images from patient line 47Q) retinal cells showed positive staining for ATXN7 (A), cone-rod homeobox (CRX, B) and recoverin (RCVRN, C) on day 30 of the differentiation experiment (nuclei in blue). qRT-PCR results confirmed the expression of *CRX* (D), *PAX6* (E), *RCVRN* (F) and *OTX2* (G) in all five lines, compared with the iPSC lines from which they were derived. All levels shown relative to β actin. (H) Expression of 23 selected genes in SCA7 patient iPSC-derived retinal cells ($n = 2$) shown relative to unaffected control cells ($n = 1$). All levels shown relative to beta actin and unaffected control cells. The data in this figure includes the cell lines from both patient cell lines (47Q and 41Q) relative to the mean the control lines ($CON_A$ and $CON_B$). A comparison against both control lines was not deemed necessary as these were shown to have similar expression profiles. $^*p < 0.05$.

control iPSC-derived cells. The differentiated cells displayed varying expression levels of the retinal genes *CRX*, *PAX6*, *RCVRN* and *OTX2* (Fig 4D–4G).

To determine transcriptional changes in the retinal cells, retinal cell specific genes were added to the gene panel used in the NPCs. The retinal genes included: arrestin 3 (*ARR3*); cone-rod homeobox (*CRX*); guanine nucleotide binding protein (G protein); alpha trans-ducing activity polypeptide 1 (*GNAT1*); Microphthalmia-associated transcription factor (*MITF*); neural retina leucine zipper (*NRL*); orthodenticle homeobox 2 (*OTX2*); paired box 6 (*PAX6*); recoverin (*RCVRN*); rhodopsin (*RHO*) and retinal pigment epithelium-specific protein 65kDa (*RPE65*). As observed in the NPC cells, both ATXN7 ($p = 0.04$) and KAT2A ($p = 0.02$) were down regulated in the retinal cells. In addition, we also noticed expression changes in genes that appeared to be specific to the SCA7 photoreceptors. These included *GRIA2*, encoding the glutamate receptor, ionotropic, AMPA2 (GluR2) (downregulated, $p = 0.04$); and three retinal-specific genes, *OTX2* (upregulated, $p = 0.002$), *RCVRN* (downregulated, $p = 0.01$) and *RPE65* (upregulated, $p = 0.001$). Lastly, and in contrast to the SCA7 NPCs, BEX1 expression appear to be upregulated in SCA7 photoreceptors ($p = 0.0006$). The opposing BEX1 transcription changes (up in photoreceptors and down in NPCs) may be indicative of a possible differential response by the different cell types to the presence of mutant ATXN7. As with the NPCs, *ATXN7* CAG repeat alleles were confirmed using an RT-PCR (S5 Fig).

## Discussion

This study describes the generation and characterisation of the first iPSCs from the South African SCA7 patient cohort, through the transduction of patient dermal fibroblasts with Sendai virus vectors (17, 18). Furthermore, we show that we are able to differentiate these iPSCs from SCA7 into neurons and retinal photoreceptors. The SCA7 patient iPSCs generated in this study were capable of differentiation to NPC, and subsequent self-renewal for up to 30 passages, with morphological similarities to neuroepithelial cells, as has been previously reported (20). The generation of sustainable populations of neural progenitors from SCA7 patients is significant, as these cells may act as a starting point for the generation of numerous disease-affected neuronal subtypes, which can be expanded for large-scale experiments at relatively low cost. Both patient- and control- derived NPCs appeared capable of differentiating into neurons expressing βIII-Tubulin and GABA, with comparable efficiencies. This corresponds with results from previous studies of neurodegenerative conditions, including other polyQ diseases (25, 26, 33), which found no difference in differentiation potential between patient and control iPSCs. In addition, no obvious difference in the ability to generate GABA-positive processes could be detected between SCA7 and control iPSC-derived neurons.

Electrophysiological studies were carried out to establish whether there were any functional differences in the intrinsic properties of patient and control cells. The majority of cells recorded were capable of generating spiking activity including single and multiple action potentials, an indication of neuronal maturity [39]. Despite the presence of significant

differences in spiking responses between the four cell lines, we did not observe a reliable trend between the control and patient derived neurons. We do acknowledge a major limitation in these findings being significant variability across cell lines. This is likely due to unappreciated differences in culturing conditions between cell lines, which may have masked our ability to detect a reliable difference in spiking responses caused by the mutant ATXN7. Alternatively, it may be that mutant ATXN7 does not affect the spiking properties of neurons at early stages of development, suggesting that an extrinsic stressor of some kind might be required to elicit a pathological phenotype. Indeed, similar functional analysis of neurons derived from patients with SCA3, a related polyQ-repeat disorder, showed no difference between control and patient cells until neurons were excited via bath application of glutamate [33].

Whilst we acknowledge that it is not possible to make conclusive claims due to the differences between all the cell lines used, we did observe an apparent difference in resting membrane potential and input resistance between control and SCA7 patient derived neurons. Patient cells had more negative resting membrane potentials and increased input resistance compared to control cells. A more hyperpolarised membrane potential is usually associated with neuronal maturity whist the high input resistance is usually associated with neuronal immaturity. These conflicting findings will need to be further explored in future studies to provide a more conclusive understanding as to how SCA7 specific changes cause changes in neuronal maturation or drive alternative functional alterations. We can, however, postulate that the more hyperpolarised membrane potential in the patient cells may be indicative of a reduction in the excitability of neurons containing mutant ATXN7 in response to synaptic input. Reductions in Purkinje cell excitability have previously been observed in animal models of SCA1 and SCA2 [40, 41] and could represent a common functional endpoint in several polyQ disorders [42]. Future work will involve exploring the underlying mechanisms that might explain these differences in resting membrane potential and input resistance. Importantly, it will be necessary to repeat this physiological functional analysis on iPSC-derived neurons of the Purkinje cell lineage in order to fully recapitulate the cerebellar-specific elements of the disease using the recently published protocols [43, 44].

Retinal differentiation of iPSCs yielded a heterogeneous cell population after 30 days, containing a large proportion of cells expressing the photoreceptor markers CRX and RCVRN, as well as retinal cell genes *PAX6*, *OTX2*, *CRX* and *NRL* in the differentiating cells. The cells expressed varying levels of the photoreceptor genes, *RCVRN* and *RHO*, confirming that the retinal differentiation protocol was capable of producing cells expressing markers of "mature" photoreceptors, albeit at relatively low levels of efficiency.

A PCR-based assay was used to confirm the size of the *ATXN7* CAG repeat alleles in cDNA from patient- and control-derived fibroblasts, iPSCs, neural and retinal cells. Besides serving as a "fingerprinting" assay, confirming the identity of the lines, these results offered a semi-quantitative analysis of the expression levels of mutant and wildtype *ATXN7* alleles, confirming that both alleles were expressed in all cell lines. Despite the inherent instability of the CAG repeat within the *ATXN7* gene [45], automated fluorescent genotyping of the PCR products indicated that no contractions or large-scale expansions had occurred during either reprogramming or differentiation, consistent with previous reports of other iPSC-derived models of polyQ disorders, including SCA3 and HD [32, 33].

Immunocytochemical analysis of ATXN7 expression in SCA7 patient neural and retinal cells showed diffuse expression within the nucleus (neurons) or nucleus and cytoplasm (retinal photoreceptors). No obvious aggregates were observed in either the iPSCs or the differentiated cells (Figs 1E, 1F and 4A). This strongly suggests that the differentiated cells represent a population of cells at an early stage of development, rather than recapitulating the age or disease stage of the patient from which the primary cells were derived. Previous studies employing

similar models for the study of neurodegenerative disease have raised concerns regarding the relevance of modelling adolescent- and adult-onset diseases over the short lifespan of cultured neurons [46, 47]. These findings suggest that pathological hallmarks of disease such as the formation of aggregates may take decades to manifest, requiring the gradual accumulation of toxic proteins as a result of age-dependent deficiencies in protein homeostasis. Although some studies suggest that aggregates may be detected at earlier stages, the major determinants of aggregate formation remain the length of the polyQ expansion, and the levels of expression of the polyQ-containing protein [48, 49]. Thus, a cell line derived from an individual expressing endogenous levels of a moderately expanded ATXN7 protein may be less likely to demonstrate an observable cellular phenotype. Alternatively, the aggregation of mutant protein may require prolonged periods in culture, or an exogenous trigger, such as exposure to oxidative stress or neurotoxins, or excitation-induced calcium influx [33].

The role of transcriptional dysregulation in the polyQ diseases has been extensively documented, particularly in cell and animal models of SCA7 [5, 6, 30, 31, 50, 51]. The identification of gene expression changes, which precede the onset of symptoms, suggests strongly that alterations in transcription may be among the earliest manifestations of disease [52]. Thus, it follows that gene expression changes may be used as a tool to identify a disease-associated phenotype in cells representing early stages of development [53].

In order to investigate gene expression changes in the SCA7 iPSCs and iPSC-derived neurons generated here, a panel of candidate transcripts were selected, in which robust changes had been previously demonstrated [4, 6, 31, 54–56]. The iPSC-derived NPCs and retinal photoreceptors displayed changes in expression of these key transcripts, suggesting that these cells may serve as useful models of neurodegenerative disease progression and for the testing of potential therapies (Figs 1D and 4H). It should be noted that the low cell numbers and heterogeneity of mature neuronal cultures proved a challenge to obtaining significant biological material to perform reliable expression analysis. Hence, the decision was made to perform this analysis on the relatively homogeneous NPC cultures, with the view that these cells may yield insights into the early cellular dysfunction underlying SCA7 pathology.

Of the three genes consistently downregulated across both differentiated cell types, two (*ATXN7* and *KAT2A*) encode components of the STAGA transcriptional co-activator complex. Previous studies in SCA7 patient fibroblasts and mouse models have demonstrated a disease-associated increase in *ATXN7* expression, mediated by non-coding RNAs [54, 57]. The contradictory decrease in *ATXN7* expression in SCA7 NPCs and photoreceptors observed here could reflect the early developmental stage of the cells, but further analysis of the regulatory pathways will be required in order to elucidate the basis for this apparent decrease in the disease-causing protein in affected cell types. *KAT2A* encodes the histone acetyltransferase GCN5. Although there are no established links between *ATXN7* and the expression of *KAT2A*, numerous studies have identified a functional interaction between the two proteins, which results in changes in STAGA activity in *in vitro* and *in vivo* models of SCA7 [4, 58, 59]. Loss of GCN5 expression has been shown to result in increased retinal degeneration in SCA7 mice [58].

The interaction between ATXN7 and CRX has been hypothesised to be a key factor behind the development of retinal degeneration in SCA7 patients [51]. Therefore, the expression of multiple known CRX targets, which were previously shown to be down-regulated in SCA7 mice, were included in the gene expression experiments. None of these target genes (including *ARR3*, *GNAT1* or *RHO*) showed consistent changes in patient cells. However, transcriptional changes in the expression of additional retinal genes, including *OTX2* (involved in the determination of photoreceptor cell fate), *RCVRN* (expressed in photoreceptors), and *RPE65* (expressed in retinal pigment epithelial cells), were noted in the patient derived cells. A

significant degree of heterogeneity was observed in the differentiated retinal cells, both in terms of morphology, and gene/protein expression (Fig 4), therefore additional investigation will be required on additional iPSC cell lines to determine whether these differences can be attributed to experimental differences or pathogenic mechanisms.

Downregulation of the HSP genes *HSP70* and *DNAJA1* was observed in SCA7 patient NPCs. A decrease in levels of these two *HSP*s has been previously reported in both SCA7 mice, and human patient lymphoblasts [6, 31]. Although this decrease in expression was hypothesised in mice to represent an advanced stage of disease progression, the early developmental stage recapitulated by our model suggests that decreases in certain HSP genes may instead be an inherent defect, which could predispose certain populations of cells to degeneration.

The generation of patient-specific, disease-relevant cell types is particularly important in neurodegenerative diseases; as such cells provide a unique model in which to evaluate disease pathogenesis without the complications associated with transgene overexpression in cell or animal models. In addition, the use of cells containing the patient's own genetic background offers the opportunity to investigate potential modifiers of disease onset and progression [60, 61]. Perhaps most importantly to the South African context, iPSC-derived neurons provide the first opportunity to evaluate the efficacy of the allele-specific RNAi-based therapy developed by Scholefield et al. [16], in disease-affected cells.

One significant caveat of this study remains the small number of patients assessed—a consequence of the rare nature of the condition—and the challenges associated with patient recruitment in a developing world setting, in which many of those affected are unable to access tertiary healthcare. Whilst, in this study, comparisons were not made between isogenic gene-edited lines, they do retain significantly similar background genetics, as they are all generated from the same immediate family, minimising (to a degree) the extensive differences between unrelated individuals. Future studies need to will focus on recruitment, in order to extend these investigations in a larger patient cohort and to allow for a greater number of cell lines to be produced. To control for the inherent genetic variability associated with comparisons between unrelated patients, future work should include the generation of isogenic control lines, by means of CRISPR/Cas9-mediated genome editing. This approach has already been demonstrated in editing trinucleotide repeat expansions in both Huntington's disease [62, 63] and myotonic dystrophy [64]. It remains, however, technically challenging to target the disease-causing repeat in ataxin-7 and retain the endogenous regulatory landscape. Due to the sequence homology of the wild -type allele, such a strategy would likely render the wild-type protein non-functional via a frameshift.

Nevertheless, the SCA7 iPSCs generated here serve as a resource for differentiation into a variety of disease-associated cell types, providing an ideal model in which to study neurodegenerative diseases. The results of this study provide evidence of a disease phenotype in iPSC-derived cells from the South African SCA7 patient cohort. We hope that our data will contribute to the ongoing efforts to use iPSC cells to study the pathogenesis of neurodegenerative disorders, which are needed in development of population-specific therapies.

## Supporting information

**S1 Fig. Genogram of SCA7 and control patients.** Genograms illustrating two families from which the SCA7 patient samples were taken from. A related control, sibling of 41Q, was used. From the three patients (two SCA7 and one control), 4 iPSC clones where derived (2 SCA7 lines and 2 Control lines).
(TIF)

**S2 Fig. Representative karyogram from a iPSC line.** Karyogram from iPSC clone P1 showing no gross abnormalities.
(TIF)

**S3 Fig. iPSC characterisation.** (A) Representative immunocytochemistry image showing positive OCT4 staining (red, top panel) and TRA-1-60 staining (green, bottom panel) in iPSCs clone 47Q. (B), Immunocytochemistry in newly infected fibroblasts (top panel) and iPSC colonies (bottom panel) co-stained with primary antibodies against OCT4 (red) and the viral nucleocapsid protein (green) showed effective silencing of the reprogramming Sendai virus. DAPI staining shown in blue. Expression of pluripotency markers in iPSC lines, determined by qRT-PCR. All five iPSC lines (47Q, 41Q, $CON_A$, $CON_B$) expressed OCT4 (C), SOX2 (D) and NANOG (E), compared to low expression levels in the original donor fibroblasts (F), or cells subjected to the retinal differentiation protocol for 10 days (D10, pooled data from lines 41Q and $CON_B$). All levels shown relative to beta (β)-actin.
(TIF)

**S4 Fig. iPSCs differentiation.** iPSCs were differentiated to three germ layers *in vitro and* validated by staining with appropriate markers: mesoderm using smooth muscle actin (SMA), ectoderm using β-III-tubulin, and endoderm using alpa-fetoprotein (AFP). Nuclei are counterstained with DAPI (blue). Images are representative of iPSCs from the 47Q patient clone.
(TIF)

**S5 Fig. Quantitative PCR results from SCA7 derived fibroblasts.** There were no significant transcriptional changes in SCA7 patient-derived fibroblasts compared to unaffected control fibroblasts.
(TIF)

**S6 Fig. Gel from semi-quantitative PCR of iPSC-derived NPCs and retinal cells.** Gel electrophoresis after qRT-PCR on NPCs (A) and retinal cells (B). Samples from all three individuals showed a single band at approximately 355bp, corresponding with a wild-type allele, whilst patient cell lines 47Q and 41Q showed an additional larger band corresponding to a mutant allele. CAG repeat length in NPCs was evaluated at varying passages (indicated by p13, p15 or p17).
(TIF)

**S1 Raw images.**
(PDF)

**S1 Table. Primary antibodies.**
(DOCX)

**S2 Table. Secondary antibodies.**
(DOCX)

**S3 Table. Primer sequences.**
(DOCX)

## Acknowledgments

The authors thank Dr Mahito Nakanishi for kind provision of the Sendai virus vector, Ms Jane Vowles for assistance with iPSC generation, and Ms Theresa Ruppelt and colleagues at the South African National Health Laboratory Service for karyotype analyses. We honour the late

Ms Ingrid Baumgarten for her invaluable assistance over many years, particularly in establishing patient fibroblast cultures.

## Author Contributions

**Conceptualization:** Lauren M. Watson, Danielle C. Smith, Susan H. Kidson, Leslie J. Greenberg.

**Data curation:** Richard J. Burman, Lauren M. Watson, Danielle C. Smith, Joseph V. Raimondo, Robea Ballo, Janine Scholefield.

**Formal analysis:** Richard J. Burman, Lauren M. Watson, Danielle C. Smith, Joseph V. Raimondo.

**Funding acquisition:** Joseph V. Raimondo, Susan H. Kidson, Leslie J. Greenberg.

**Investigation:** Richard J. Burman, Lauren M. Watson, Danielle C. Smith, Joseph V. Raimondo, Robea Ballo.

**Methodology:** Richard J. Burman, Lauren M. Watson, Danielle C. Smith, Robea Ballo.

**Project administration:** Richard J. Burman, Danielle C. Smith, Leslie J. Greenberg.

**Supervision:** Joseph V. Raimondo, Sally A. Cowley, Matthew J. A. Wood, Susan H. Kidson, Leslie J. Greenberg.

**Writing – original draft:** Richard J. Burman, Lauren M. Watson, Danielle C. Smith, Joseph V. Raimondo, Susan H. Kidson, Leslie J. Greenberg.

**Writing – review & editing:** Richard J. Burman, Lauren M. Watson, Danielle C. Smith, Joseph V. Raimondo, Robea Ballo, Janine Scholefield, Sally A. Cowley, Matthew J. A. Wood, Susan H. Kidson, Leslie J. Greenberg.

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
