## [Decision Letter · Decision Letter 0]

14 Oct 2020

Pécs, Hungary

October 13, 2020

PONE-D-20-27215

Molecular and electrophysiological features of spinocerebellar ataxia type seven in induced pluripotent stem cells

PLOS ONE

Dear Dr. Burman,

Thank you for submitting your manuscript to PLOS ONE. After careful consideration, we feel that it has merit but does not fully meet PLOS ONE’s publication criteria as it currently stands. Therefore, we invite you to submit a revised version of the manuscript that addresses the points raised by the Reviewers, listed below.

We look forward to receiving your revised manuscript.

Kind regards,

Joseph Najbauer, Ph.D.

Academic Editor

PLOS ONE

Journal Requirements:

'Ethics approval for the study was granted by the University of Cape Town (UCT) Faculty of Health Sciences Human Research Ethics Committee (HREC REF. 380/2009 and 434/2011), and was renewed annually, incorporating amendments to the project protocol where necessary. All methods were carried out in accordance with the guidelines approved by the Ethics Committee. Participants were recruited from the Neurogenetics clinic at Groote Schuur Hospital in Cape Town. Informed consent was obtained from all participants prior to their enrolment in the study.'

Please provide additional details regarding participant consent. In the ethics statement in the Methods and online submission information, please ensure that you have specified what type you obtained (for instance, written or verbal, and if verbal, how it was documented and witnessed). If your study included minors, state whether you obtained consent from parents or guardians. If the need for consent was waived by the ethics committee, please include this information.

'Funding for this work was provided by Ataxia UK, Commonwealth Scholarship Commission (UK), John Fell OUP Fund, National Research Foundation (South Africa), National Research Foundation (South Africa, Competitive Programme for Rated Researchers CPR20110624000019696), Medical Research Council (South Africa), Harry Crossley Foundation, Deutscher Akademischer Austausch Dienst, University of Cape Town Research Council, Wellcome Trust, Parkinson’s Disease UK, Medical Research Council UK, Blue Brain Project, and the James Martin 21st Century School.'

'The funders had no role in study design, data collection and analysis, decision to

publish, or preparation of the manuscript.'

Please include the updated Funding Statement in your cover letter. We will change the online submission form on your behalf.

Reviewers' comments:

Reviewer's Responses to Questions

**Comments to the Author**

1. Is the manuscript technically sound, and do the data support the conclusions?

Reviewer #1: Partly

Reviewer #2: Partly

Reviewer #3: Yes

Reviewer #4: Yes

Reviewer #5: No

2. Has the statistical analysis been performed appropriately and rigorously? 

Reviewer #1: No

Reviewer #2: Yes

Reviewer #3: I Don't Know

Reviewer #4: Yes

Reviewer #5: Yes

3. Have the authors made all data underlying the findings in their manuscript fully available?

Reviewer #1: No

Reviewer #2: Yes

Reviewer #3: Yes

Reviewer #4: Yes

Reviewer #5: Yes

4. Is the manuscript presented in an intelligible fashion and written in standard English?

Reviewer #1: Yes

Reviewer #2: Yes

Reviewer #3: Yes

Reviewer #4: Yes

Reviewer #5: Yes

5. Review Comments to the Author

Reviewer #1: The authors describe the electrofisiological and molecular features in iPS derivated cells derived from 2 SCA7 patients and 1 control. Although, the results could be interesting the manuscript contains a lot of errors and lacks some fundamental data and results.

1. Errors in references. In the page 7 the authors cite the reference 16 what is not their previous article and is not related to the text.

2. Material and methods. Retinal differentiation lack some fundamental details. It is not clear until which day of differentiation toward retinal cells, the cells were maintained.

3. The Supplementary Fig 2 should include the sequencing results to confirm the the disease genotype

4. Page 7. The authors claim” the reprogramming Sendai virus, were confirmed by immunocytochemistry (antibodies listed in S1-2 Tables). The expression of selected pluripotency genes (OCT4, SOX2, NANOG) was determined by quantitative PCR.

All results related to confirmation of silencing of the virus (RT-PCR is normally used) and expression of the endogenous pluripotent genes should be shown in manuscript.

Also the shape of the colonies. The alkaline phosphate assay is missing of undifferentiated cells. Also panel should include differentiation capacity to three embryonic layers at least for 2 markers for each layer. And this all for 3 generated lines.

Fingerprint analysis of generated lines is missing. Other pluripotency markers are missing for immune in S3 Figure.

S3 B Hoecht staining is of low quality for the first figure.

Immunocytochemistry has to be shown for all three lines (the best clon of each)

It is not clear what C1, C2 and others mean. Different clones of the same patient and Control lines???

It is strange that C2 and P2b does not express OCT2. Are they pluripotent? It is not clear why the authors continu the differentiation with iPS lines with no expression of pluripotent markers. Or the authors did not include tha data to confirm a full pluripotency of the generated lines.

No data which representative iPS line is shown in S3

The data is lacking.

5. Results: Fig 1B the immune staining is low quality. The cells have no neuronal forms.

Fig 1C It is strange that expression of PAX6 is so low and it is early neural marker.

Fig 1 D n=2 for these kind of data is too low.

6. Fig4. All cells are recoverin positive. Normally this protocol generates small yield of recoverin positive cells. It is no clear how many baches of differentiation protocol were used for statistical analysis.

In conclusion, the manuscript does not have sufficient technical level for publication in PLOS ONE. Experiments, statistics, and other analyses are not performed to a high technical standard and are not described in sufficient detail

Reviewer #2: Manuscript by Burman, Watson, Smith et al. is a very interesting and valuable study of SCA7 pathology in patient-derived cell lines. These cell lines include iPSCs, reprogrammed from human fibroblasts, as well as cell lines obtained by neuronal differentiation. Neuronal cells were assessed for electrophysiological properties and examined for transcriptional aberrations. Cell lines were characterized comprehensively, both iPSCs and neuronal cell lines.

My major concern is about the number of cell lines generated and analyzed. I do understand that generation and characterization of this kind of cell lines is difficult and often troublemaking. But the issue here is whether the use of this number of cell lines is enough for conclusions made. In numerous studies isogenic cell lines are used, which is more proper approach when only small number of cell lines can be used (due to complicated experimental setup etc.). I would find the results more convincing if the experiments were performed for more cell lines. For example the results presented in the Figure 3E, although the tendency of change is not consistent for SCA7 cell lines, is shown as statistically significant lowering of the measured parameter in Fig. 3F. The same refers to results presented in Fig. 3 G and H.

First of all, it is not quite clear from the lines 107-108 and 127-128 how many control cell lines were generated. I see in Fig. S1 that two clones of control cell line were analyzed, but lines 127-128 could be rephrased to state this clearly.

For results presented in Fig. 1D, n is 2 for control samples, but actually two clones of the same cell line were used. Were the results very consistent for these clones? If so treating them as two control cell lines is a bit misleading. If no – why?

Lines 316-317: Again naming two clones of iPSC line as “two separated iPSC lines” is misleading.

Other major comments:

- Images in figure 1F for SCA7 and control cell line do not look like performed with identical settings. Also for better comparison they should be at the same magnification.

- I cannot see reference to Fig. 1F in the text, whereas reference to Fig. 1E is only in the discussion. The same refers to Fig. 2 A-D. This is confusing.

Minor comments:

- Line 119: “quantitative RT-PCR” should be mentioned (RT – reverse transcription), instead of “quantitative PCR” (and throughout the text: RT-qPCR or qRT-PCR, instead of qPCR)

- I found some essential references missing, like recent review papers: PMID: 31792895, 31432449; as well as experimental ones: 27999335, 31859031

- Table S1: it is valuable also to mention catalogue numbers as some suppliers have several antibodies for specific protein

- Fig. 1A: green word “NESTIN” is hardly visible when placed at the image. The same refers to Fig. 4 A-C.

- Line 406: two patients can be named a “cohort”?

Reviewer #3: Burman et al present the novel generation of patient-derived iPSC models of SCA7. Differentiation of iPSCs to neurons and photoreceptors enabled the finding of electrophysiological immaturity in SCA7 neurons and transcriptional alterations resulting from ATXN7 repeat expansions.

The work is original and well carried out and the manuscript is very well written. I am supportive of its publication in PLOS ONE, with some minor consideration for the authors.

1) Please can you clarify the number of batches of retinal cells used in Fig 4H (line 386). Is this 2 batches of patient cells relative to 1 control batch, or are both clones represented? I would say that two batches are minimum for this type of analysis.

2) Please can you expand upon the iPSC culture conditions. Line 114. KOSR percentage, bFGF concentration. How were the colonies manually passaged?

3) Line 150 – how many days of induction was performed before NSCs were produced? How many passages were NPCs cultured for before neural induction. Was this consistent between controls and patients? Neural and glial potential can shift with prolonged NSC passaging and maybe this is worth mentioning in the discussion?

4) Consider a statistics section. I am unsure why Fig 2E used Chi-squared, Fig 3C used Kruskal Wallis and Fig 3E/G used ANOVA.

Minor

Line 342 please correct iPS to iPSC

Line 365 please correct sentence, e.g. is “we asked” missing?

Line 427 please correct sentence, e.g. is “are significant” missing?

Reviewer #4: The manuscript by Burman et al. assesses molecular and electrophysiological features of neurons and retinal photoreceptor cells derived from induced pluripotent stem cells (iPSCs) that were reprogrammed from fibroblasts from two unrelated patients suffering from spinocerebellar ataxia type seven (SCA-7), and compared them with iPSC-derived cells from one healthy control donor. Authors report transcriptional aberrations in SCA-7 patient derived neural stem cells as well as photoreceptor cells, and differences in resting membrane potential and input resistance between control and SCA-7 patient derived neurons. Although the study is mainly descriptive due to the low number of subjects (2 patients and 1 healthy control), it is scientifically sound and the results are relevant for future studies on iPSC-based phenotyping of neurodegenerative diseases.

I recommend resolving only two very minor issues before publication:

1. The references to some of the figures in the main text is incorrect, and – in those cases – does not correspond to the figure legends (lines 280, 282, 294, 295).

2. The sentence in line 365 on p. 18 is incomplete.

Reviewer #5: Within the manuscript Burman et al. generated iPSCs from two SCA7 patients and one control, differentiated iPSCs into iPSC-derived neurons and photoreceptors and analysed the molecular and electrophysiological properties of these cells.

Unfortunately, the inaccurate and incomplete experimental setup does not allow to draw any final conclusions.

This has mainly two major reasons:

- The line to line variability independent from the disease background is to high to only use 2 control lines (derived from the same individual) and 2 patient lines. The authors claim that SCA7 patients show electrophysiological differences, although the electrophysiological results are variable and inconclusive towards a maturation phenotype.

Timing of electrical maturation can greatly vary among different iPSC lines. Therefore, in order to compare control to diseased neurons, analysis of multiple electrophysiological parameters might be more predictive. To confirm a developmental functional phenotype analysis of multiple control and patient lines especially at different time points during the differentiation is absolutely essential.

- the characterisation of differentiated cells (neurons and retinal photoreceptors) is too inaccurate. Thereforem it is unclear if line to line differences only occur due to different compositions of cell types and/or different differentiation stages (cultivation artefact).

Detailed comments:

Electrophysiology:

- In general there are nearly no differences between the control (C1a / C1b) and the matching patient line (P2b)

Inconsistent data:

298 - the spiking response of a cell to current injection can be used to determine the maturation stage of a differentiating neuron

424 - Despite the presence of significant differences in spiking responses between the four cell lines, we did not observe a reliable trend between the control and patient derived neurons.

- No differences of SCA7 neurons in neuronal maturity as determined by the occurrence of single and multiple action potentials (Fig 2E)

310 - The resting membrane potentials (Vm) of cells varied significantly across each cell line. The Vm derived from SCA7 patient lines was significantly more hyperpolarised compared to the control cell lines.

- No explanation within the text on how resting membrane potential is relevant or indicative for impaired neuronal health or maturation.

- Generally, increasingly hyperpolarized resting membrane potentials are a hallmark of maturation. In this respect, pooled SCA7 patient lines show a more mature phenotype.

332 - Next, we performed voltage-clamp recordings of the neurons in order to measure the input resistance as well as the voltage-gated sodium and potassium currents (Fig 3A-B). A lower input resistance is associated with neurite outgrowth and increased numbers of ion channels inserted into the plasma membrane during the process of neuronal maturation. (…). Cells derived from SCA7 patient lines had a significantly higher mean input resistance than cells from the control lines (Fig 3D, p = 0.01, Mann-Whitney U test).

- Generally, neurons change developmentally with a decrease in input resistance. In this respect, pooled SCA7 patient lines show a more immature phenotype

341 - (Fig 3) Sodium and potassium currents in neurons differentiated from control and patient iPS cell lines

360 - When pooled together, however, we noted that both the Max IK and Max INa were significantly smaller in the cell lines coming from SCA7 patients compared to the control cell lines.

- High variance between patient lines, only significantly different from controls when pooled together

- Increasing max sodium and potassium currents reflect the process of electrical maturation. In this regard, SCA7 patient lines show a more immature phenotype

Expression analysis:

- Why were only NPCs used for the analysis in Fig. 1D? What about iPSC-derived neurons?

- It is unclear how the qPCR results were normalized (two control lines?)

- High variance in Fig. 4 D-G between the two control lines. How was this normalized in Fig. 4H?

General comments:

- In depth characterisation of iPSCs is missing (Suppl. Fig.)

- Where multiple housekeeping genes used for qPCR quantification?

- p15, l282: Fig 2G-H: data not shown, Fig. 1?

- The type of generated neurons is unclear (any marker? glutamatergic, GABAergic, regional identity?)

- Which lines were used for the representative images of the ICC stainings?

- p19, ll 375-376: Obvious difference: The expression of CRX in control and SCA7 cells seems to be different (Fig. 4B)

6. PLOS authors have the option to publish the peer review history of their article (what does this mean?). If published, this will include your full peer review and any attached files.

Reviewer #1: No

Reviewer #2: No

Reviewer #3: No

Reviewer #4: No

Reviewer #5: No

---

## [Author Response · Author response to Decision Letter 0]

4 Dec 2020

We thank the Editor and Reviewers for their thoughtful comments and suggestions. In the rebuttal below, we have addressed all the points with our responses shown in italics. We have also highlighted all changes to the original manuscript in yellow. 

Journal Requirements:

Response: Noted, we have formatted our manuscript according to the PLoS ONE guidelines provided. 

'Ethics approval for the study was granted by the University of Cape Town (UCT) Faculty of Health Sciences Human Research Ethics Committee (HREC REF. 380/2009 and 434/2011), and was renewed annually, incorporating amendments to the project protocol where necessary. All methods were carried out in accordance with the guidelines approved by the Ethics Committee. Participants were recruited from the Neurogenetics clinic at Groote Schuur Hospital in Cape Town. Informed consent was obtained from all participants prior to their enrolment in the study.'

Please provide additional details regarding participant consent. In the ethics statement in the Methods and online submission information, please ensure that you have specified what type you obtained (for instance, written or verbal, and if verbal, how it was documented and witnessed). If your study included minors, state whether you obtained consent from parents or guardians. If the need for consent was waived by the ethics committee, please include this information.

Response: The ethics statement has been amended to state that written informed consent was obtained by all participants.

'Funding for this work was provided by Ataxia UK, Commonwealth Scholarship Commission (UK), John Fell OUP Fund, National Research Foundation (South Africa), National Research Foundation (South Africa, Competitive Programme for Rated Researchers CPR20110624000019696), Medical Research Council (South Africa), Harry Crossley Foundation, Deutscher Akademischer Austausch Dienst, University of Cape Town Research Council, Wellcome Trust, Parkinson’s Disease UK, Medical Research Council UK, Blue Brain Project, and the James Martin 21st Century School.'

'The funders had no role in study design, data collection and analysis, decision to

publish, or preparation of the manuscript.'

Please include the updated Funding Statement in your cover letter. We will change the online submission form on your behalf.

Response: Noted, any funding information has been removed from the Acknowledgements. The Funding Statement is included in the cover letter and can be updated as follows:

‘Funding for this work was provided by Ataxia UK, Commonwealth Scholarship Commission (UK), John Fell OUP Fund, National Research Foundation (South Africa), National Research Foundation (South Africa, Competitive Programme for Rated Researchers CPR20110624000019696), Medical Research Council (South Africa), Harry Crossley Foundation, Deutscher Akademischer Austausch Dienst, University of Cape Town Research Council, Wellcome Trust, Parkinson’s Disease UK, Medical Research Council UK, Blue Brain Project, and the James Martin 21st Century School. Electrophysiology equipment was provided by the École Polytechnique Fédérale de Lausanne, Switzerland. The funders had no role in study design, data collection and analysis, decision to publish, or preparation of the manuscript.’

 

Reviewer #1: 

Reviewer: The authors describe the electrofisiological and molecular features in iPS derivated cells derived from 2 SCA7 patients and 1 control. Although, the results could be interesting the manuscript contains a lot of errors and lacks some fundamental data and results.

Response: We thank the reviewer for the effort put into appraising our manuscript. 

Reviewer: Errors in references. In the page 7 the authors cite the reference 16 what is not their previous article and is not related to the text.

Response: Reference 16 is a general reference describing the establishment of fibroblast cultures from biopsy. The position of the reference in the text has been altered to reflect this.

Reviewer: Material and methods. Retinal differentiation lack some fundamental details. It is not clear until which day of differentiation toward retinal cells, the cells were maintained. 

Response: The text has been amended to state that the retinal differentiation period was 30 days.

Reviewer: The Supplementary Fig 2 should include the sequencing results to confirm the the disease genotype

Response: CAG repeat sizing results are presented in Supplementary Figure 5.

Reviewer: Page 7. The authors claim” the reprogramming Sendai virus, were confirmed by immunocytochemistry (antibodies listed in S1-2 Tables). The expression of selected pluripotency genes (OCT4, SOX2, NANOG) was determined by quantitative PCR.

All results related to confirmation of silencing of the virus (RT-PCR is normally used) and expression of the endogenous pluripotent genes should be shown in manuscript. Also the shape of the colonies. The alkaline phosphate assay is missing of undifferentiated cells. Also panel should include differentiation capacity to three embryonic layers at least for 2 markers for each layer. And this all for 3 generated lines. Fingerprint analysis of generated lines is missing. Other pluripotency markers are missing for immune in S3 Figure. S3 B Hoecht staining is of low quality for the first figure. Immunocytochemistry has to be shown for all three lines (the best clon of each) It is not clear what C1, C2 and others mean. Different clones of the same patient and Control lines??? It is strange that C2 and P2b does not express OCT2. Are they pluripotent? It is not clear why the authors continu the differentiation with iPS lines with no expression of pluripotent markers. Or the authors did not include tha data to confirm a full pluripotency of the generated lines. No data which representative iPS line is shown in S3. The data is lacking.

Response: We acknowledge that not all pluripotency techniques available were performed, however with clarification below we hope to alleviate the reviewers’ concerns to prove that the necessary characterizations were completed. In certain sections for this question we first refer to the reviewer’s comment and respond in italics:

Reviewer: “All results related to confirmation of silencing of the virus (RT-PCR is normally used) and expression of the endogenous pluripotent genes should be shown in manuscript.”

Response: As the reviewer correctly indicates, one method of confirming the absence of non-integrating episomal Sendai virus can be shown by RT-PCR. However, this can equally be shown by IF; i.e. “This clearance rate is clone-dependent and can be confirmed by PCR or by anti-Sendai antibody” (manufacturer’s instructions and as a published example of prior art, Lee et al. 2015). Hence, representative images of clone X are shown in S3 Fig (see panel B). 

Expression of endogenous pluripotent genes as measured by qRT-PCR in each iPSC clone is shown in S3 Fig panels C-E. These are shown for both control clones (CONA and CONB) and the two patient clonal lines (47Q and 41Q). Expression of pluripotent genes as measured by IF evaluation of protein expression were further performed. Representative images are shown in S3 Fig panel A. Fingerprint analysis of iPSC lines is often used for confirmation that the final cell models are derived from the correct individual. 

References:

Link to manufacturer's instructions:

https://tools.thermofisher.com/content/sfs/brochures/CytoTune-iPS%202%200%20Sendai%20Reprogramming%20Kit%20FAQs.pdf

Lee HK, Morin P, Wells J, Hanlon EB, Xia W. Induced pluripotent stem cells (iPSCs) derived from frontotemporal dementia patient's peripheral blood mononuclear cells. Stem Cell Research. 2015; 15(2): 325-27. doi: 10.1016/j.scr.2015.07.004

Reviewer: “Also panel should include differentiation capacity to three embryonic layers at least for 2 markers for each layer.”

Response: We thank the reviewer for pointing out the omissions. Whilst we did not complete double labelling, we followed prior art as shown in other publications (such as Fig 1C in Hartfield et al. 2014) and have inserted a new supplementary figure and added a caption to the manuscript (see S4 Fig). 

Reference: 

Hartfield EM, Yamasaki-Mann M, Ribeiro Fernandes HJ, Vowles J, James WS, Cowley SA, et al. (2014) Physiological Characterisation of Human iPS-Derived Dopaminergic Neurons. PLoS ONE 9(2): e87388. https://doi.org/10.1371/journal.pone.0087388

Reviewer: “Different clones of the same patient and Control lines???

Response: We agree that the naming of the iPSC lines could be misleading and therefore have renamed them as follows: 

PSCs from an affected patient and her affected daughter were used to generate two patient derived lines from each respective individual:

47Q

41Q

iPSCs from an unaffected sibling genetically confirmed to have a normal allele size control was used to generate two clones:

CONA

CONB 

This has been corrected in the text and the figures and is specifically highlighted in S1 Fig. 

As per the methods section, “Primary dermal fibroblast cultures were established from punch skin biopsies (16) taken from the inner forearm of two unrelated SCA7 patients (41Q and 47Q) and an unaffected control individual (CONA and CONB, sibling of 41Q)”. As we had samples from two SCA7 patients but only dermal fibroblasts from one control individual we generated two clones from the control to minimise differences in epigenetic reprogramming 

Reviewer: “It is strange that C2 and P2b does not express OCT2. Are they pluripotent? It is not clear why the authors continu the differentiation with iPS lines with no expression of pluripotent markers.”

Response: As per our previous comment above, all the iPSC clones generated do express OCT4 (which we believe the reviewer is speaking to) to show pluripotency in S3 Fig in addition to the additional pluripotency data we show as outlined above. If we understand the reviewer correctly, they may be referring to Fig 1C where we show that neural progenitors from these clones no longer express OCT4 to indicate they have lost pluripotency and are moving towards PAX6 neural progenitor cells as indicated in the figure legend. 

Reviewer: “No data which representative iPS line is shown in S3.”

Response: iPSC clone P1 (now 47Q) is shown as a representative image in S5 Fig. 

Reviewer: Fig 1B the immune staining is low quality. The cells have no neuronal forms.

Fig 1C It is strange that expression of PAX6 is so low and it is early neural marker.

Fig 1 D n=2 for these kind of data is too low.

Response: These are not fully differentiated neurons, and more closely resemble neuroepithelial cells in culture. In each case, n = 2 biological replicates, performed in technical triplicate. The text has been amended to reflect this.

Reviewer: Fig4. All cells are recoverin positive. Normally this protocol generates small yield of recoverin positive cells. It is no clear how many baches of differentiation protocol were used for statistical analysis.

Response: The methods section has been updated to include the following description: 

“Each patient and control line underwent two rounds of differentiation, with each time point analysed in biological duplicate.”

 

Reviewer #2: 

Reviewer: Manuscript by Burman, Watson, Smith et al. is a very interesting and valuable study of SCA7 pathology in patient-derived cell lines. These cell lines include iPSCs, reprogrammed from human fibroblasts, as well as cell lines obtained by neuronal differentiation. Neuronal cells were assessed for electrophysiological properties and examined for transcriptional aberrations. Cell lines were characterized comprehensively, both iPSCs and neuronal cell lines.

Response: We thank the reviewer for the thoughtful and encouraging comments. 

Reviewer: My major concern is about the number of cell lines generated and analyzed. I do understand that generation and characterization of this kind of cell lines is difficult and often troublemaking. But the issue here is whether the use of this number of cell lines is enough for conclusions made. In numerous studies isogenic cell lines are used, which is more proper approach when only small number of cell lines can be used (due to complicated experimental setup etc.). I would find the results more convincing if the experiments were performed for more cell lines. For example the results presented in the Figure 3E, although the tendency of change is not consistent for SCA7 cell lines, is shown as statistically significant lowering of the measured parameter in Fig. 3F. The same refers to results presented in Fig. 3 G and H.

Response: The reviewer raises an important point. Indeed, we considered this as a strategy, however generating isogenic lines from CAG triplet repeat disorders is extremely challenging. For example, Dastidar et al (2018) were able to excise the CTG repeat expansion in myotonic dystrophy patient-derived iPSCs. Due to the sequence homology between the wild-type and mutant alleles, the subsequent cells were shown to include CRISPR-mediated targets on the wild-type allele. Whilst this did not impact the protein coding region of the DMPK gene because of the 3’UTR location of the triplet repeat, the same strategy targeting the disease-causing repeat in ataxin-7 would likely render the wild-type protein non-functional via a frameshift. Furthermore, the precise correction of 42 to 12 CAG repeats is incredibly technically challenging given the repetitive nature required of the repair template. Thus we were restricted to patient derived iPSC lines. Whilst these are not isogenic they do retain significantly similar background genetics, as they are all generated from the same immediate family, minimising (to a degree) the extensive differences between unrelated individuals. In addition, whilst SCA7 patients are more common in South Africa, this remains a rare genetic disease and we were limited to two patients. 

To acknowledge the reviewer’ concern we have added the following to the text in the discussion, “It remains, however, technically challenging to target the disease-causing repeat in ataxin-7 and retain the endogenous regulatory landscape. Due to the sequence homology of the wild -type allele, such a strategy would likely render the wild-type protein non-functional via a frameshift.”

Reviewer: First of all, it is not quite clear from the lines 107-108 and 127-128 how many control cell lines were generated. I see in Fig. S1 that two clones of control cell line were analyzed, but lines 127-128 could be rephrased to state this clearly.

Response: We have amended the methods section lines 116 and 117 to clarify this and to ensure it is clear throughout the manuscript. 

For results presented in Fig. 1D, n is 2 for control samples, but actually two clones of the same cell line were used. Were the results very consistent for these clones? If so treating them as two control cell lines is a bit misleading. If no – why?

Lines 316-317: Again, naming two clones of iPSC line as “two separated iPSC lines” is misleading.

Response: We have amended the methods section to clarify this and to ensure it is clear throughout the manuscript. 

Reviewer: Images in figure 1F for SCA7 and control cell line do not look like performed with identical settings. Also for better comparison they should be at the same magnification.

Response: We thank the reviewer for noting this. We have corrected the image to include the same area imaged with the correct (same) magnification.

Reviewer: I cannot see reference to Fig. 1F in the text, whereas reference to Fig. 1E is only in the discussion. The same refers to Fig. 2 A-D. This is confusing.

Response: We amended the text to ensure all figures are referenced correctly. 

Reviewer: Line 119: “quantitative RT-PCR” should be mentioned (RT – reverse transcription), instead of “quantitative PCR” (and throughout the text: RT-qPCR or qRT-PCR, instead of qPCR).

Response: The text has been edited accordingly.

Reviewer: I found some essential references missing, like recent review papers: PMID: 31792895, 31432449; as well as experimental ones: 27999335, 31859031 

Response: We thank the reviewer for making us aware of these references. We have included them in the following section of the introduction: 

“More recently, dysfunction in cell metabolism has been proposed as an additional pathogenic mechanism (8, 9). 

As with many neurodegenerative conditions, research into the molecular pathogenesis of SCA7 has been hindered by a lack of suitable models of human disease progression. This is particularly relevant in cases where the genomic context of the mutation may have an impact on gene function and might prove useful for therapeutic development. There does, however, seem to be increasing momentum in this area with new models being proposed and refined (10-12).”

Reviewer: Table S1: it is valuable also to mention catalogue numbers as some suppliers have several antibodies for specific protein

Response: These have been added. 

Reviewer: Fig. 1A: green word “NESTIN” is hardly visible when placed at the image. The same refers to Fig. 4 A-C.

Response: We have adjusted the colours of the labels to a lighter shade so that they stand more clearly against the darker images. 

Reviewer: Line 406: two patients can be named a “cohort”?

Response: We agree with the Reviewer that the term ‘cohort’ is inappropriate when including only two patients. In the context of our study, however, we refer the South African cohort as being a group of more than two patients who have been diagnosed with SCA7. From this cohort of patients, only two patients were available and willing to participate in this study. 

 

Reviewer #3: 

Burman et al present the novel generation of patient-derived iPSC models of SCA7. Differentiation of iPSCs to neurons and photoreceptors enabled the finding of electrophysiological immaturity in SCA7 neurons and transcriptional alterations resulting from ATXN7 repeat expansions.

The work is original and well carried out and the manuscript is very well written. I am supportive of its publication in PLOS ONE, with some minor consideration for the authors.

Response: We thank the reviewer for the encouraging remarks. 

Reviewer: Please can you clarify the number of batches of retinal cells used in Fig 4H (line 386). Is this 2 batches of patient cells relative to 1 control batch, or are both clones represented? I would say that two batches are minimum for this type of analysis.

Response: The data in this figure includes the cell lines from both patients (P1 and P2b now changed to 47Q and 41Q) relative to one of the control lines (C1a now CONA). A comparison against both control lines was not deemed necessary as there were shown to have similar expression profiles. We have included this clarification in the figure caption. 

Reviewer: Please can you expand upon the iPSC culture conditions. Line 114. KOSR percentage, bFGF concentration. How were the colonies manually passaged?

Response: We have corrected the following in the text: 

One week after infection, the cells were transferred to mitomycin C-inactivated feeder cells in stem cell medium (KO DMEM, 20% KOSR, 1% NEAA, 50 µM β-mercaptoethanol, glutamax, 10ng/ml bFGF). Half of the medium was refreshed every day and colonies were expanded by manual passaging (dissection of colonies using needles) on feeder layers as previously described (Schwartz et al. 2011).

Reviewer: Line 150 – how many days of induction was performed before NSCs were produced? How many passages were NPCs cultured for before neural induction. Was this consistent between controls and patients? Neural and glial potential can shift with prolonged NSC passaging and maybe this is worth mentioning in the discussion?

Response: NPCs were observed in culture after seven days of differentiation, and neurite outgrowth was also typically observed after approximately one week of switching NPCs to neuronal differentiation media. The text has been edited to reflect this. Every attempt was made to treat patient and control lines consistently, to ensure accurate comparisons could be made.

Reviewer: Consider a statistics section. I am unsure why Fig 2E used Chi-squared, Fig 3C used Kruskal Wallis and Fig 3E/G used ANOVA.

Response: We have included a more detailed description on the choice of statistical tests in the ‘Electrophysiology’ part of the Methods section. To answers the Reviewer’s comments directly: 

1) In Fig. 2E a Chi-squared test was used to assess for differences in the proportion or fraction (%) of cells in each category of cell firing across the different cell lines. 

2) In Fig. 3C we realise that we incorrectly wrote Kruskal Wallis instead of ANOVA. This was a typographical error and we confirm that an ANOVA test was used. We have corrected this in the text. The ANOVA is the statistical test of choice as it allows us to identify differences across all cell lines. 

Minor:

Line 342 please correct iPS to iPSC 

Line 365 please correct sentence, e.g. is “we asked” missing?

Line 427 please correct sentence, e.g. is “are significant” missing?

Response: We thank the reviewer and have corrected these minor comments. 

 

Reviewer #4: 

The manuscript by Burman et al. assesses molecular and electrophysiological features of neurons and retinal photoreceptor cells derived from induced pluripotent stem cells (iPSCs) that were reprogrammed from fibroblasts from two unrelated patients suffering from spinocerebellar ataxia type seven (SCA-7), and compared them with iPSC-derived cells from one healthy control donor. Authors report transcriptional aberrations in SCA-7 patient derived neural stem cells as well as photoreceptor cells, and differences in resting membrane potential and input resistance between control and SCA-7 patient derived neurons. Although the study is mainly descriptive due to the low number of subjects (2 patients and 1 healthy control), it is scientifically sound and the results are relevant for future studies on iPSC-based phenotyping of neurodegenerative diseases. 

I recommend resolving only two very minor issues before publication: 

Reviewer: The references to some of the figures in the main text is incorrect, and – in those cases – does not correspond to the figure legends (lines 280, 282, 294, 295).

Response: We have corrected the text to ensure all figures are referenced correctly. 

Reviewer: The sentence in line 365 on p. 18 is incomplete.

Response: We thank the reviewer and have corrected.

 

Reviewer #5: 

Within the manuscript Burman et al. generated iPSCs from two SCA7 patients and one control, differentiated iPSCs into iPSC-derived neurons and photoreceptors and analysed the molecular and electrophysiological properties of these cells. Unfortunately, the inaccurate and incomplete experimental setup does not allow to draw any final conclusions.

Response: We thank the reviewer for the effort put into appraising our manuscript. 

This has mainly two major reasons:

The line to line variability independent from the disease background is to high to only use 2 control lines (derived from the same individual) and 2 patient lines. The authors claim that SCA7 patients show electrophysiological differences, although the electrophysiological results are variable and inconclusive towards a maturation phenotype. Timing of electrical maturation can greatly vary among different iPSC lines. Therefore, in order to compare control to diseased neurons, analysis of multiple electrophysiological parameters might be more predictive. To confirm a developmental functional phenotype analysis of multiple control and patient lines especially at different time points during the differentiation is absolutely essential.

The characterisation of differentiated cells (neurons and retinal photoreceptors) is too inaccurate. Therefore, it is unclear if line to line differences only occur due to different compositions of cell types and/or different differentiation stages (cultivation artefact).

Response: We thank the reviewer for these comments. We accept and fully acknowledge that the limited number of cell lines and large variability make it difficult to make firm conclusions from our data. We are completely transparent about this and have aimed to address these issues to the best of our ability and within our limited capacity. We do, however, believe that our study does provide useful experimental insights to those who are working in this field. 

Detailed comments:

Electrophysiology: 

Reviewer: In general there are nearly no differences between the control (C1a / C1b) and the matching patient line (P2b)

Response: We acknowledge that there are limited difference between the patient and control lines. We are transparent about this and explain that this is likely due to the limited number of cell lines. 

Reviewer: 298 - the spiking response of a cell to current injection can be used to determine the maturation stage of a differentiating neuron

Response: We followed a similar approach used by other studies which use spiking response to assess neuronal maturation in different stem cell lines. Specific references include: 

Moore AR, Filipovic R, Mo Z, Rasband MN, Zecevic N, Antic SD. Electrical excitability of early neurons in the human cerebral cortex during the second trimester of gestation. Cerebral Cortex. 2008;19(8):1795-805.

Perrier AL, Tabar V, Barberi T, Rubio ME, Bruses J, Topf N, et al. Derivation of midbrain dopamine neurons from human embryonic stem cells. Proceedings of the National Academy of Sciences of the United States of America. 2004;101(34):12543-8.

Vazin T, Becker KG, Chen J, Spivak CE, Lupica CR, Zhang Y, et al. A novel combination of factors, termed SPIE, which promotes dopaminergic neuron differentiation from human embryonic stem cells. PloS one. 2009;4(8):e6606.

Belinsky GS, Moore AR, Short SM, Rich MT, Antic SD. Physiological properties of neurons derived from human embryonic stem cells using a dibutyryl cyclic AMP-based protocol. Stem cells and development. 2011;20(10):1733-46.

Reviewer: 424 - Despite the presence of significant differences in spiking responses between the four cell lines, we did not observe a reliable trend between the control and patient derived neurons.

Response: As above, we are transparent about the limited differences between the control and patient derived neurons we observed. 

Reviewer: No differences of SCA7 neurons in neuronal maturity as determined by the occurrence of single and multiple action potentials (Fig 2E)

Response: As above, we are transparent about the limited differences between the control and patient derived neurons. 

Reviewer: 310 - The resting membrane potentials (Vm) of cells varied significantly across each cell line. The Vm derived from SCA7 patient lines was significantly more hyperpolarised compared to the control cell lines.

Response: We acknowledge that there was significantly variability across the different cell lines with this being more apparent in some parameters compared to others. 

Reviewer: No explanation within the text on how resting membrane potential is relevant or indicative for impaired neuronal health or maturation. Generally, increasingly hyperpolarized resting membrane potentials are a hallmark of maturation. In this respect, pooled SCA7 patient lines show a more mature phenotype.332 - Next, we performed voltage-clamp recordings of the neurons in order to measure the input resistance as well as the voltage-gated sodium and potassium currents (Fig 3A-B). A lower input resistance is associated with neurite outgrowth and increased numbers of ion channels inserted into the plasma membrane during the process of neuronal maturation. (…). Cells derived from SCA7 patient lines had a significantly higher mean input resistance than cells from the control lines (Fig 3D, p = 0.01, Mann-Whitney U test). Generally, neurons change developmentally with a decrease in input resistance. In this respect, pooled SCA7 patient lines show a more immature phenotype

Response: A neuron’s resting membrane potential (Vm) affects its excitability. Neurons with more hyperpolarised potentials are less excitable (i.e. less likely to respond to synaptic inputs and fire action potentials). Disease processes have been associated with either depolarised (Jeub et al. 2015) or hyperpolarised Vm (Kiernan et al. 2002). Typically, however, a more hyperpolarised Vm is associated with neurons that are more mature. To highlight this point, we have added the following to our discussion: 

“A more hyperpolarised membrane potential is usually associated with neuronal maturity whist the high input resistance is usually associated with neuronal immaturity. These conflicting findings will need to be further explored in future studies to provide a more conclusive understanding as to how SCA7 specific changes cause changes in neuronal maturation or drive alternative functional alterations. We can, however, postulate that the more hyperpolarised membrane potential in the patient cells may be indicative of a reduction in the excitability of neurons containing mutant ATXN7 in response to synaptic input.”

References:

Jeub M, Herbst M, Spauschus A, Fleischer H, Klockgether T, Wuellner U, Evert BO. Potassium channel dysfunction and depolarized resting membrane potential in a cell model of SCA3. Experimental neurology. 2006;201(1):182-92. doi: https://doi.org/10.1016/j.expneurol.2006.03.029

Kiernan MC, Guglielmi JM, Kaji R, Murray NM, Bostock H. Evidence for axonal membrane hyperpolarization in multifocal motor neuropathy with conduction block. Brain. 2002 Mar 1;125(3):664-75. doi: https://doi.org/10.1093/brain/awf041

Reviewer: 341 - (Fig 3) Sodium and potassium currents in neurons differentiated from control and patient iPS cell lines. 360 - When pooled together, however, we noted that both the Max IK and Max INa were significantly smaller in the cell lines coming from SCA7 patients compared to the control cell lines. High variance between patient lines, only significantly different from controls when pooled together

Response: We acknowledge that there are significant differences between patient lines and for this reason the pooled data (control vs SCA7) is misleading. For this reason, we have opted to remove the pooled data from the figures and text. We now report the cell lines individually and report on observed trends whilst highlighting that these would need be further explored in future studies. 

Reviewer: Increasing max sodium and potassium currents reflect the process of electrical maturation. In this regard, SCA7 patient lines show a more immature phenotype

Response: We are hesitant to draw strong conclusions from this data given the variability of the patient lines. 

Expression analysis:

Reviewer: Why were only NPCs used for the analysis in Fig. 1D? What about iPSC-derived neurons?

Response: The low cell numbers and heterogeneity of mature neuronal cultures proved a challenge to obtaining significant biological material to perform reliable expression analysis. Hence, the decision was made to perform this analysis on the relatively homogeneous NPC cultures, with the view that these cells may yield insights into the early cellular dysfunction underlying SCA7 pathology.

Reviewer: It is unclear how the qPCR results were normalized (two control lines?)

Response: Results are shown relative to the mean of the two control lines (set as 1), and normalised to the housekeeping gene, β-actin.

Reviewer: High variance in Fig. 4 D-G between the two control lines. How was this normalized in Fig. 4H?

Response: As above, Results are shown relative to the mean of the two control lines (set as 1), and normalised to the housekeeping gene, β-actin.

General comments:

Reviewer: In depth characterisation of iPSCs is missing (Suppl. Fig.)

Response: We followed prior art as shown in other publications (such as Figure 1C in Hartfield et al. 2014) and have inserted a new supplementary figure (see S3 Fig) 

Reference: 

Hartfield EM, Yamasaki-Mann M, Ribeiro Fernandes HJ, Vowles J, James WS, Cowley SA, et al. (2014) Physiological Characterisation of Human iPS-Derived Dopaminergic Neurons. PLoS ONE 9(2): e87388. https://doi.org/10.1371/journal.pone.0087388

Reviewer: Where multiple housekeeping genes used for qPCR quantification? 

Response: Only β-actin was used as a housekeeping gene, as its levels were observed not to vary according to disease state. 

Reviewer: p15, l282: Fig 2G-H: data not shown, Fig. 1?

Response: This was a typographical error and has been corrected. The figures are now correctly referenced. 

Reviewer: The type of generated neurons is unclear (any marker? glutamatergic, GABAergic, regional identity?)

Response: Fig 1F shows that a proportion of the neurons stain positive for GABA. However, as this was intended as a pilot study on a small number of patients and controls, extensive phenotyping of the neuronal population was not conducted.

Reviewer: Which lines were used for the representative images of the ICC stainings?

Response: For Fig 1, CONA and 41Q were used as representative control and patient lines, respectively. Pluripotency staining images in S4 Fig. used CONB.

Reviewer: p19, ll 375-376: Obvious difference: The expression of CRX in control and SCA7 cells seems to be different (Fig 4B)

Response: Whilst representative images have been selected for the figure, no consistent differences in expression were noted across the differentiation experiments.

---

## [Decision Letter · Decision Letter 1]

30 Dec 2020

Pécs, Hungary

December 28, 2020

PONE-D-20-27215R1

Molecular and electrophysiological features of spinocerebellar ataxia type seven in induced pluripotent stem cells

PLOS ONE

Dear Dr. Burman,

Thank you for submitting your manuscript (R1 version) to PLOS ONE. After careful consideration, we feel that it has merit but does not fully meet PLOS ONE’s publication criteria as it currently stands. Therefore, we invite you to submit a revised version of the manuscript that addresses the points raised by Reviewer #2 and Reviewer #5, listed below.

We look forward to receiving your revised manuscript.

Kind regards,

Joseph Najbauer, Ph.D.

Academic Editor

PLOS ONE

Reviewers' comments:

Reviewer's Responses to Questions

**Comments to the Author**

1. If the authors have adequately addressed your comments raised in a previous round of review and you feel that this manuscript is now acceptable for publication, you may indicate that here to bypass the “Comments to the Author” section, enter your conflict of interest statement in the “Confidential to Editor” section, and submit your "Accept" recommendation.

Reviewer #1: All comments have been addressed

Reviewer #2: (No Response)

Reviewer #3: All comments have been addressed

Reviewer #4: All comments have been addressed

Reviewer #5: (No Response)

2. Is the manuscript technically sound, and do the data support the conclusions?

Reviewer #1: Yes

Reviewer #2: Partly

Reviewer #3: Yes

Reviewer #4: Yes

Reviewer #5: No

3. Has the statistical analysis been performed appropriately and rigorously? 

Reviewer #1: Yes

Reviewer #2: Yes

Reviewer #3: Yes

Reviewer #4: Yes

Reviewer #5: No

4. Have the authors made all data underlying the findings in their manuscript fully available?

Reviewer #1: Yes

Reviewer #2: No

Reviewer #3: Yes

Reviewer #4: Yes

Reviewer #5: Yes

5. Is the manuscript presented in an intelligible fashion and written in standard English?

Reviewer #1: Yes

Reviewer #2: Yes

Reviewer #3: Yes

Reviewer #4: Yes

Reviewer #5: Yes

6. Review Comments to the Author

Reviewer #1: The authors addressed properly all my concerns. They replied correctly.

Reviewer #2: The majority of my comments was sufficiently addressed, except for the comment concerning interpretation of the results based on two SCA7 cell lines analyzed.

First of all, I am not convinced by explanation for not generating isogenic cell lines by comparison with DMPK gene. I know that the CRISPR/Cas-9-based strategy is difficult to be applied for repeated tracts and its application for this study would take a long time. Still, there are studies like PMID: 28238795 and 32182692 which show application of such strategy for huntingtin, where CAG repeat tract is also present in ORF. It can be referred and commented instead.

Unfortunately, in attached files I could not access current version of Fig. 3. (zip files contained only main figures 1 and 4) which was changed as I see in this figure legend and the interpretation.

Reviewer #3: The authors have satisfactorily addressed my comments.

I recommend that this manuscript is appropriate for publication.

Reviewer #4: (No Response)

Reviewer #5: Unfortunately the revised manuscript "Molecular and electrophysiological features of spinocerebellar ataxia type seven in induced pluripotent stem cells" does not lead to any improvement in either of the two main critical points.

1.) Since the line to line variability is too high, no conclusive results can be obtained.

2.) Even more important, the identity of differentiated cell types (either neurons or retinal photoreceptors) is still more or less unknown. As the authors stated in their response: "The low cell numbers and heterogeneity of mature neuronal cultures

proved a challenge to obtaining significant biological material to perform reliable expression analysis." If even such an analysis was not possible, then the results of the electrophysiological recordings are inappropriate. The maturity and the cellular composition/identity of differentiated neurons determines the electrophysiological properties. Therefore, it can not be excluded that the "features of functional aberrations" in SCA7 iPSC-derived neurons are a cell culture artefact rather than a consequence of the mutation.

Without any additional experiments to prove the identity of differentiated neurons (proportion of GABAergic vs. glutamatergic neurons / neuronal subtype specification) and photoreceptors no conclusive findings can be formulated.

7. PLOS authors have the option to publish the peer review history of their article (what does this mean?). If published, this will include your full peer review and any attached files.

Reviewer #1: No

Reviewer #2: No

Reviewer #3: No

Reviewer #4: No

Reviewer #5: No

---

## [Author Response · Author response to Decision Letter 1]

16 Jan 2021

We thank the Editor and Reviewers for their thoughtful comments and suggestions. In the rebuttal below, we have addressed all the points with our responses shown in italics. We have also highlighted all changes to the original manuscript in yellow. 

Reviewer #2:

Reviewer: The majority of my comments was sufficiently addressed, except for the comment concerning interpretation of the results based on two SCA7 cell lines analyzed.

First of all, I am not convinced by explanation for not generating isogenic cell lines by comparison with DMPK gene. I know that the CRISPR/Cas-9-based strategy is difficult to be applied for repeated tracts and its application for this study would take a long time. Still, there are studies like PMID: 28238795 and 32182692 which show application of such strategy for huntingtin, where CAG repeat tract is also present in ORF. It can be referred and commented instead.

Response: We thank the reviewer for these comments and agree with what has been suggested. We have revised the manuscript to expand our discussion on future experiments and the application of CRISPR/Cas-9-based strategies (see lines 548-557). We also thank the Reviewer for the references they have suggested which we have now included in our manuscript. 

Reviewer: Unfortunately, in attached files I could not access current version of Fig. 3. (zip files contained only main figures 1 and 4) which was changed as I see in this figure legend and the interpretation. 

Response: We apologise to the Reviewer for this technical glitch. We can confirm that all the Figures were successfully uploaded during the resubmission. We have ensured that all Figures are again uploaded successfully upon submission of this revision. 

Reviewer #5:

Reviewer: Unfortunately the revised manuscript "Molecular and electrophysiological features of spinocerebellar ataxia type seven in induced pluripotent stem cells" does not lead to any improvement in either of the two main critical points: 

1.) Since the line to line variability is too high, no conclusive results can be obtained.

2.) Even more important, the identity of differentiated cell types (either neurons or retinal photoreceptors) is still more or less unknown. As the authors stated in their response: "The low cell numbers and heterogeneity of mature neuronal cultures

proved a challenge to obtaining significant biological material to perform reliable expression analysis." If even such an analysis was not possible, then the results of the electrophysiological recordings are inappropriate. The maturity and the cellular composition/identity of differentiated neurons determines the electrophysiological properties. Therefore, it cannot be excluded that the "features of functional aberrations" in SCA7 iPSC-derived neurons are a cell culture artefact rather than a consequence of the mutation. Without any additional experiments to prove the identity of differentiated neurons (proportion of GABAergic vs. glutamatergic neurons / neuronal subtype specification) and photoreceptors no conclusive findings can be formulated. 

Response: We thank the Reviewer for their critical appraisal of our study. We agree with the Reviewer that given the significant variability between cell lines, it is not appropriate to make conclusive statements from the findings. We have, however, avoided from doing so and have instead aimed to present our results as they are whilst fully acknowledging the limitations. In this revision, we have ensured that the tone of our manuscript does not mislead the reader into thinking that our findings are conclusive and have again highlighted the limitations. In addition, we clearly highlight what future research is needed to take the work forward. 

We do believe our paper does still provide technical insights that may prove useful to others especially those considering embarking on developing or refining iPSC models of SCA7. While we agree that the experiments you suggest to prove the identity of the differentiated cell are warranted, at present this is not possible. This is largely due to the significant negative impact the ongoing COVID-19 pandemic has had on our research efforts as well as the ongoing restrictions it is having on our capacity.

---

## [Decision Letter · Decision Letter 2]

8 Feb 2021

Pécs, Hungary

February 7, 2021

Molecular and electrophysiological features of spinocerebellar ataxia type seven in induced pluripotent stem cells

PONE-D-20-27215R2

Dear Dr. Burman,

We’re pleased to inform you that your manuscript (R2 version) has been judged scientifically suitable for publication and will be formally accepted for publication once it meets all outstanding technical requirements.

Kind regards,

Joseph Najbauer, Ph.D.

Academic Editor

PLOS ONE

Reviewers' comments:

Reviewer's Responses to Questions

**Comments to the Author**

1. If the authors have adequately addressed your comments raised in a previous round of review and you feel that this manuscript is now acceptable for publication, you may indicate that here to bypass the “Comments to the Author” section, enter your conflict of interest statement in the “Confidential to Editor” section, and submit your "Accept" recommendation.

Reviewer #2: All comments have been addressed

Reviewer #5: (No Response)

2. Is the manuscript technically sound, and do the data support the conclusions?

Reviewer #2: Yes

Reviewer #5: Yes

3. Has the statistical analysis been performed appropriately and rigorously? 

Reviewer #2: Yes

Reviewer #5: Yes

4. Have the authors made all data underlying the findings in their manuscript fully available?

Reviewer #2: Yes

Reviewer #5: Yes

5. Is the manuscript presented in an intelligible fashion and written in standard English?

Reviewer #2: Yes

Reviewer #5: Yes

6. Review Comments to the Author

Reviewer #2: (No Response)

Reviewer #5: I fully understand that under the current COVID 19-crisis it is difficult to perform the necessary experiments to address my two main points of criticism (line to line variability, missing characterization of differentiated cells).

I still believe that analyzing patients to controls without knowing the identity and potential differences of the differentiated cell type is a major error which makes any interpretation of the existing data difficult.

Nevertheless this study may be a good starting point for the establishment of a patient-derived model to investigate pathogenic mechanism in SCA7.

7. PLOS authors have the option to publish the peer review history of their article (what does this mean?). If published, this will include your full peer review and any attached files.

Reviewer #2: No

Reviewer #5: No

---

## [Editor Report · Acceptance letter]

15 Feb 2021

PONE-D-20-27215R2 

Molecular and electrophysiological features of spinocerebellar ataxia type seven in induced pluripotent stem cells 

Dear Dr. Burman:

I'm pleased to inform you that your manuscript has been deemed suitable for publication in PLOS ONE. Congratulations! Your manuscript is now with our production department. 

Kind regards, 

on behalf of

Dr. Joseph Najbauer 

Academic Editor

PLOS ONE